# Electroencephalographic response to transient adaptation of vestibular perception

Josephine I. Cooke[1], Onur Guven[1], Patricia Castro Abarca[1,2], Richard T. Ibitoye[1] 🆔, Vito E. Pettorossi[3] and Adolfo M. Bronstein[1] 🆔

[1] *Neuro-otology Unit, Department of Brain Sciences, Imperial College London, Charing Cross Hospital, London, UK*
[2] *Escuela de Fonoaudiología, Facultad de Medicina, Clínica Alemana Universidad del Desarrollo, Santiago, Chile*
[3] *Dipartimento di Medicina e Chirurgia, Sezione di Fisiologia Umana e Biochimica, Università Degli Studi di Perugia, Perugia, Italy*

Handling Editors: Richard Carson & Jing-Ning Zhu

The peer review history is available in the Supporting Information section of this article (https://doi.org/10.1113/JP282470#support-information-section).

**Abstract** When given a series of sinusoidal oscillations in which the two hemicycles have equal amplitude but asymmetric velocity, healthy subjects lose perception of the slower hemicycle (SHC), reporting a drift towards the faster hemicycle (FHC). This response is not reflected in

the vestibular–ocular reflex, suggesting that the adaptation is of higher order. This study aimed to define EEG correlates of this adaptive response. Twenty-five subjects underwent a series of symmetric or asymmetric oscillations and reported their perceived head orientation at the end using landmarks in the testing room; this was converted into total position error (TPE). Thirty-two channel EEG was recorded before, during and after adaptation. Spectral power and coherence were calculated for the alpha, beta, delta and theta frequency bands. Linear mixed models were used to determine a region-by-condition effect of the adaptation. TPE was significantly greater in the asymmetric condition and reported error was always in the direction of the FHC. Regardless of condition, alpha desynchronised in response to stimulation, then rebounded back toward baseline values. This pattern was accelerated and attenuated in the prefrontal and occipital regions, respectively, in the asymmetric condition. Functional connectivity networks were identified in the beta and delta frequency bands; these networks, primarily comprising frontoparietal connections, were more coherent during asymmetric stimulation. These findings suggest that the temporary vestibulo-perceptual 'neglect' induced by asymmetric vestibular stimulation may be mediated by alpha rhythms and frontoparietal attentional networks. The results presented further our understanding of brain rhythms and cortical networks involved in vestibular perception and adaptation.

(Received 22 November 2021; accepted after revision 6 June 2022; first published online 17 June 2022)

**Corresponding author** Adolfo M. Bronstein: Neuro-otology Unit, Department of Brain Sciences, Imperial College London, Charing Cross Hospital, Fulham Palace Rd, London W6 8RF, UK. Email: a.bronstein@imperial.ac.uk

**Abstract figure legend** Three-panelled figure showing an overview of study methods and major findings. Left: side view graphic of rotating chair that was used to deliver asymmetric, whole-body rotation around an Earth vertical axis. Below in blue is a sample stimulus profile of the asymmetric chair oscillations (not to scale). The bottom diagram indicates the result of such stimulation, namely that repeated asymmetric oscillations lead to a loss of perception of the slower movement (indicated by red 'X'). Centre: two representative head-plots showing normalised power at different time points over a series of asymmetric oscillations. At bottom, a general data trend indicating how alpha power changed over time, with different rates of change shown for occipital and prefrontal electrodes. Right: axial view of scalp, with a simplified (illustrative) connectivity network overlayed.

### Key points

- Whole-body asymmetric sinusoidal oscillations, which consist of hemicycles with equal amplitude but differing velocities, can induce transient 'neglect' of the slower hemicycle in the vestibular perception of healthy subjects.
- In this study, we aimed to elucidate EEG correlates of this 'neglect', thereby identifying a cortical role in vestibular perception and adaptation.
- We identified a desynchronisation–resynchronisation response in the alpha frequency band (8–14 Hz) that was accelerated in the prefrontal region and attenuated in the occipital region when exposed to asymmetric, as compared to symmetric, rotations.
- We additionally identified functional connectivity networks in the beta (14–30 Hz) and delta (1–4 Hz) frequency bands consisting primarily of frontoparietal connections.
- These results suggest a prominent role of alpha rhythms and frontoparietal attentional networks in vestibular perception and adaptation.

## Introduction

It was shown by Pettorossi et al. (2013) that perceptive and reflexive vestibular mechanisms have different adaptive responses to prolonged whole-body motion around an Earth-vertical axis. Asymmetric stimulation, in which a subject experiences fast rotation to one side and slow rotation to the other, resulted in a rapid adaptation and perceptual 'neglect' of the slower movement, reflected in a substantial error when reporting perceived position. As a result, subjects experience a rapid perceptual shift in the spatial representation of the body-facing direction toward the slower movement, as well as only perceiving rotation in the direction of the faster movement. These perceptual changes occur quickly and persist for several

minutes after the rotations end. This response was not observed in vestibulo-ocular reflex (VOR) responses (Faralli et al., 2022; Panichi, Faralli et al., 2017; Panichi, Occhigrossi et al., 2017; Pettorossi et al., 2013), suggesting a higher-order central mechanism might underpin the perceptual adaptation.

There has been little research investigating EEG markers of vestibular adaptation; much of the existing work has investigated EEG during visuospatial and motor adaptation. These studies suggest that a frontoparietal attentional network mediates error processing during adaptation (Tan et al., 2014; Torrecillos et al., 2015). This network can be further split into a dorsal and ventral network, which mediate top-down goal-oriented behaviour and bottom-up attentional reorienting, respectively (Corbetta & Shulman, 2002; Petersen & Posner, 2012; Scolari et al., 2015). A high beta frequency band has been linked to the dorsal network, in which signals originate in the frontal cortex then move posteriorly to the parietal cortex; the ventral network is linked to signals which originate in the parietal cortex then move forward and is mediated by a low gamma frequency band (Buschman & Miller, 2007).

Alpha frequencies have been implicated in adaptive processes for visuospatial, motor and vestibular tasks; it has been shown that alpha desynchronises in response to sensory stimulation, and then rebounds toward baseline values (Gale et al., 2016; Savoie et al., 2018; Tan et al., 2014). This pattern is also seen in the beta band and is localised bilaterally over the parietal region when the stimulus is vestibular in nature (Gale et al., 2016). Despite this existing research, neurophysiological correlates of purely vestibular adaptation remain under-investigated, but research in this area is necessary, given that clinical work suggests a prominent role for the cerebral hemispheres in adapting to dizziness after vestibular lesions (Cousins et al., 2017; Dieterich & Brandt, 2008; Panichi, Faralli et al., 2017).

Thus, this study sought to further clarify EEG correlates of vestibular perception and how it adapts over various tasks, using the asymmetric rotation paradigm (Pettorossi et al., 2013). Our aim was to identify and characterise EEG markers of vestibulo-perceptual adaption during asymmetric rotation. We hypothesised that the pattern of desynchronisation and subsequent resynchronisation discussed above would be disturbed with asymmetric rotation (during the adaptation period), especially in the alpha and beta bands over parieto-occipital regions, as found previously (Gale et al., 2016; Gutteling & Medendorp, 2016). Additionally, we hypothesised that frontoparietal networks would be activated to monitor increased perceptual error and resolve potential conflicts in incoming information during the asymmetric rotation task. This hypothesis derives from the theory postulated by Pettorossi et al. (2013) that, during rotations combining high and low frequencies, subjects could 'neglect' slow components of the oscillation and focus on more functionally meaningful high frequencies. Additionally, if the asymmetric adaptation process represents a focal reduction of activity in cortical vestibular areas, EEG changes should be more prominent in the posterior insular and parietal regions theorised to be involved in vestibular processing (Lopez & Blanke, 2011).

## Methods

### Ethical approval

Written, informed consent was obtained from all individuals before their participation in the study. The study protocol was approved by the local research ethics committee (North East- – York Research Ethics Committee: 17/NE/0133) and the experiment was performed in accordance with the ethical standards set out in the *Declaration of Helsinki*, except for registration in a database.

### Subjects

Twenty-five young, right-handed participants [12 male, age range: 19–37 years old, mean age (SD): 28.24 (6.31) years old] volunteered for this experiment. All participants had normal or corrected vision and had no history of neurological or labyrinthine dysfunction.

### Experimental design

The experiments took place inside a Faraday cage, with participants seated on a rotating chair (Toennies Nystagliner, Höchberg, Germany) around the yaw axis, thus stimulating the horizontal semi-circular canals. To reduce the influence of non-vestibular sensory cues, head and body movement was restrained with padded clamps and adjustable belts. During the experiment, the room was darkened, though not completely so the experimenter could navigate within the room; participants were required to close their eyes during trials to fully remove visual input, although they were tasked to maintain focus as if they were looking straight ahead. Ear plugs were used to minimise possible auditory cues.

Four stimulation profiles were constructed, consisting of 16 sinusoidal cycles with a 40° total amplitude (Fig. 1*A*). The velocities of each hemicycle were either matched (symmetrical stimulation) or unmatched (asymmetrical stimulation). The chair was manually operated, to reduce electrical noise contaminating the EEG signal. This was carried out by the first author, who was aided by an in-ear metronome (set to 60 bpm to count the duration

for each hemicycle) and a low-power laser (moved along a protractor affixed to the base of the chair, thus allowing the experimenter to track stimulus amplitude). The experimenter and the laser beam set-up were behind the participant.

There were two symmetrical conditions, and two asymmetrical conditions (Fig. 1*A*) resulting in a 2 × 2 experimental design. All frequencies were in the range of vestibular perception, that is, above 0.10 Hz (Fernandez & Goldberg, 1971). The low-frequency (LF) symmetrical condition consisted of two 4 s hemicycles for an overall 8 s cycle (whole cycle frequency: 0.125 Hz; hemicycle frequency: 0.25 Hz), while the high-frequency (HF) symmetrical condition consisted of two 2 s hemicycles for an overall 4 s cycle (whole cycle frequency: 0.25 Hz; hemicycle frequency: 0.5 Hz). The slow hemicycle (SHC) of the asymmetrical conditions corresponded to the hemicycle duration for the LF and HF symmetric conditions while the fast hemicycle (FHC) was always 1 s (1 Hz) in duration. Thus, the LF asymmetrical condition consisted of a 1 s FHC and a 4 s SHC for an overall 5 s cycle (whole cycle

frequency: 0.2 Hz; FHC frequency: 1 Hz; SHC frequency: 0.25 Hz), and the HF asymmetrical condition consisted of a 1 s FHC and a 2 s SHC for an overall 3 s cycle (whole cycle frequency: 0.33 Hz; FHC frequency: 1 Hz; SHC frequency: 0.5 Hz) (Fig. 1*A*). The degree of asymmetry for the asymmetric profiles was calculated as the ratio of the duration between the SHC and the duration of the whole cycle multiplied by 100; therefore, the LF asymmetrical profile was 80% asymmetric and the HF asymmetrical profile was 66.67% asymmetric.

The experimenter practised rotating the chair at these frequencies at length before beginning the experiment; traces were examined at the end of each session to ensure regularity and consistency of the stimulus. In addition, at the conclusion of the study, the recorded stimulus traces of 10 randomly selected participants were overlayed to check for discrepancies (Supplementary Fig. 1). A Fast Fourier Transform (FFT) was also performed on the traces, and the results were overlayed (Supplementary Fig. 2). Both analyses showed minimal variability in the shape and frequency content of the stimuli, thus

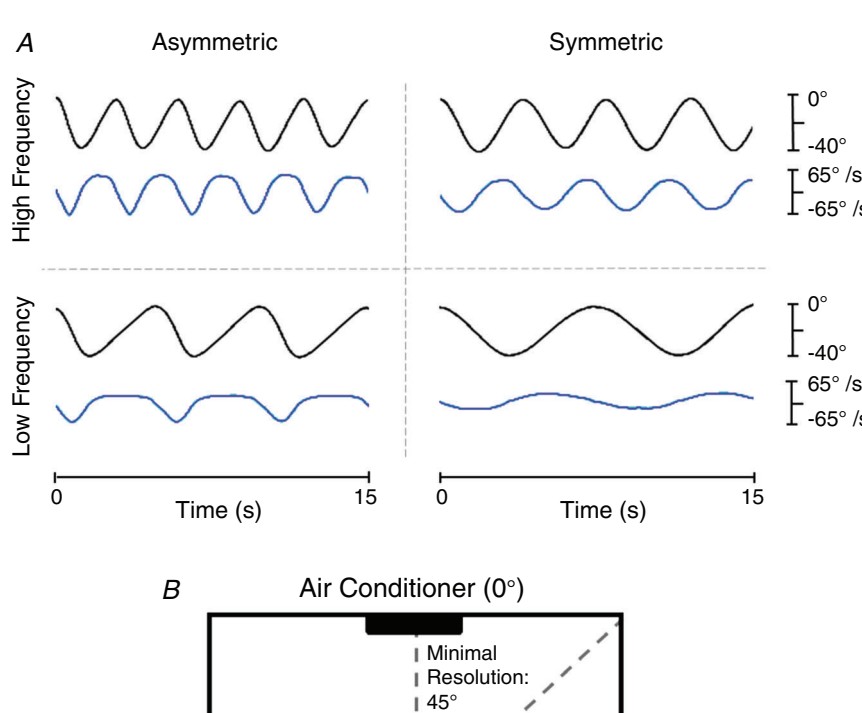

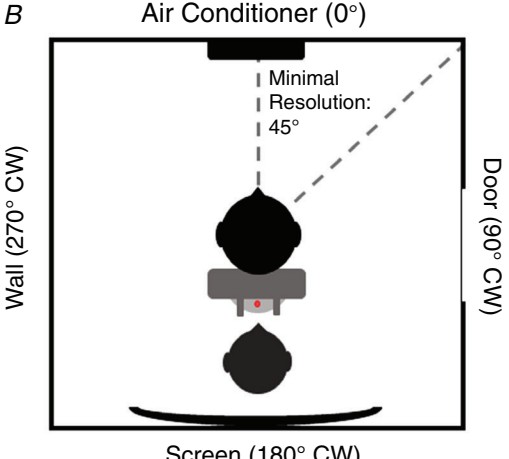

**Figure 1. Experimental stimuli and set-up**

*A*, sample traces of chair position (black) and chair velocity (blue) for each condition over a 15 s period. Negative values indicate degree of leftward displacement (position; 40° of total displacement) and leftward movement of the chair (velocity). *B*, birds-eye view of room landmarks in the Faraday cage with corresponding degree of clockwise (CW) displacement. The subject (black) sat in the rotating chair (medium grey) which was retrofitted with handles the experimenter (dark grey) used to manually operate the chair. A protractor (light grey) affixed to the base of the chair was traced with a laser (red) to monitor chair displacement. Minimal resolution for perceptual error was 45°, that is the distance between a landmark, and the corner between it and the next landmark.

demonstrating the stimulation was consistent across participants.

Due to the previous finding that asymmetric vestibular stimulation can result in altered perception for up to 30 min (Pettorossi et al., 2013), participants experienced each condition once, with the symmetric conditions always presented before the asymmetric conditions. The order of HF *vs.* LF for the symmetric and asymmetric conditions was counterbalanced to ensure no order effects. However, given the known hemispheric asymmetry of cortical vestibular processing – the right hemisphere being dominant for vestibular and spatial processing (Arshad et al., 2015; Dieterich et al., 2003) – we chose to have the FHC always move in a leftward direction, always preceding the SHC and only test right-handed participants.

### Experimental protocol

Participants were kept naïve to the paradigm and to the fact that the chair was operated manually. They were told that they would be experiencing rotations and that they should try to remain aware of where they were facing. Before the beginning of the experiment, they were directed to look at four landmarks in the Faraday cage: the air conditioner (0°; subjects always began facing this landmark and used this as a point of reference), the door (90°CW – clockwise), the screen (180°) and the blank wall (270°CW). Participants were instructed to use these reference points to describe their perceived facing after each session. The corners between any two landmarks could also be used by participants to describe their facing; as such, the minimum resolution for measuring perceptual error was 45° (Fig. 1*B*). Although this resolution was somewhat coarse, previous work (Pettorossi et al., 2013) indicated that three to four oscillations usually elicits an error in the range 30–40°, with additional cycles demonstrating a cumulative error. With participants experiencing 16 cycles, it followed that if they did not report an error of at least 45°, adaptation had not occurred. Perceptual error was quantified as the difference between the reference landmark (air conditioner – always switched off) and their perceived facing and could be greater than 360° if participants felt they made more than one full revolution.

For each session, EEG was first recorded for 2 min, without the chair moving, to establish a baseline which would later be used to normalise data recorded during and after adaptation. Following this, participants were informed that the chair would begin moving for approximately 1–2 min, whilst EEG was also recorded (*Adaptation* period). Immediately after this, there was another 2 min period in which EEG was recorded without chair movement (*Post-Adaptation* response). Finally, at the culmination of the recording session, participants

were asked to state their final perceived facing. Their response was converted to total position error (TPE), as described above.

The participants were not given feedback about the accuracy of their response and were told they would be moved at a sub-threshold frequency back to face the front. Participants opened their eyes in between each condition to verify they were facing the air conditioner and to help eliminate any after-effects of the stimulation. Every participant was debriefed after they had completed all four sessions. No adverse outcomes or persistent sensations of disorientation were reported by participants after study completion.

### EEG data acquisition

EEG was recorded with a 32-electrode scalp system (ANT Neuro, Enschede, The Netherlands) according to the 10–20 system, sampled at 1250 Hz, and referenced to a 33rd electrode (GND, position AFz) anterior to Fz. Electrode impedances were kept below 5 kΩ. As an ancillary measure to verify that participants were adapting as previously described (Faralli et al., 2022; Panichi, Faralli et al., 2017; Panichi, Occhigrossi et al., 2017; Pettorossi et al., 2013), eye movements were recorded simultaneously [bi-temporal electroocoulogram (EOG)] with electrodes placed on the lateral canthi of each eye. Note that while EOG is not conventionally recorded with eyes closed, as it increases roving eye movement artefacts, it was necessary that participants close their eyes, recalling that the room was not completely darkened so that the experimenter could move the chair accurately. Chair displacement and velocity were recorded through a potentiometer and tachometer in the base of the chair.

### EEG data pre-processing

EEG data were collected using Asa<sup>TM</sup> software (ANT Neuro, Hengelo, the Netherlands) and pre-processed using custom MATLAB code. Each recording first underwent a low-pass anti-aliasing finite impulse response (FIR) filter with a passband frequency of 100 Hz, a stopband frequency of 125 Hz, a pass-band ripple of 0.001 dB, and a stop-band attenuation of 80 dB. Signals were then down-sampled to 250 Hz and further filtered with a high-pass FIR filter (passband frequency of 1 Hz, stopband frequency of 0.8 Hz, pass-band ripple of 0.01 dB, and a stop-band attenuation of 80 dB) and another low-pass FIR filter (passband frequency of 45 Hz, stopband frequency of 49 Hz, pass-band ripple of 0.01 dB and a stop-band attenuation of 60 dB). After filtering, the data were visually examined and any epochs with electrical, EMG or eye movement artefacts were rejected. The data were then further de-noised using independent component analysis

(ICA); resultant components were visually inspected and any components (maximum of four) indicating EOG and ECG contamination were removed.

### EEG analysis

Using a FFT, steady-state spectral power was calculated for all three recording blocks (*Baseline*, *Adaptation* and *Post-Adaptation*) across four frequencies: delta (1–4 Hz), theta (4–8 Hz), alpha (8–14 Hz) and beta (14–30 Hz). For the alpha frequency band, power was measured at individual alpha frequency (IAF). The upper bound of alpha frequencies is usually defined at 12 or 13 Hz; however, IAF has been found to go up to 14 Hz (Babiloni, Barry et al., 2020; Babiloni, Noce et al., 2020; Klimesch, 1999), hence the larger band.

To reduce inter-individual variance of EEG, the *Adaptation* and *Post-Adaptation* recordings were normalised (divided) by the baseline recording. This normalisation process additionally provided an indication of the relationship between raw power during the *Adaptation* period *versus* the *Baseline* period. Specifically, it indicates if power is decreased during the *Adaptation* period relative to baseline, (note: as this reduction results from decreased synchrony of the underlying neural population, it is termed desynchronisation) or increased (increased synchrony; termed synchronisation) (Pfurtscheller, 2001; Pfurtscheller & Lopes Da Silva, 1999). Thus, normalised power values >1 indicated synchronisation, while values <1 indicated desynchronisation.

To capture time-sensitive changes (cycle-by-cycle effects), EEG recorded during the *Adaptation* period was split into epochs corresponding to each cycle of the chair. Epochs were 5 s (asymmetric) or 8 s (symmetric) in length, measured from peak displacement of each cycle of the chair position traces. EEG spectrograms, showing FFT changes as a function of time, were generated for each of these epochs to examine adaptation-specific changes in power between conditions.

Finally, average magnitude-squared coherence and imaginary coherence in each frequency band were also calculated between all electrodes for the *Baseline*, *Adaptation* and *Post-Adaptation* periods.

### Statistical analysis

Statistical analyses were all run in CRAN R version 4.0.2. Data organization and visualization was accomplished with the *tidyverse* (Wickham et al., 2019), *ggsignif* (Ahlmann-Eltze, 2019) and *rstatix* (Kassambara, 2020) packages. Outliers were defined using the boxplot method: data greater than the value of the third quartile plus 1.5 times the interquartile range were excluded from the relevant dataset. Unless otherwise specified, data were analysed using linear mixed-effects models (LMMs), which modelled both fixed-effect parameters associated with covariates of interest (brain region, experimental condition, frequency, etc.) and random effects, which model random deviance of outcome measures from fixed factors of interest, thus improving the predictive power of a model (West et al., 2015). Unlike conventional linear models, LMMs are also able to include subjects with missing data, thereby allowing maximal use of data even after outlier removal and epoch rejection. We used LMMs that included both fixed effects (region, condition, etc.) and random effects associated with the individual deviations from the fixed intercept for each condition, as well as allowing for heterogenous residual variances (West et al., 2015). This was done using the *nmle* package (Pinheiro et al., 2020). An ANOVA was then run on each model to determine the significance of main effects and interactions. When applicable, *post hoc* pairwise comparisons were carried out with the *emmeans* package (Lenth, 2020). Adjustments for multiple comparisons were made using the false discovery rate (Benjamini, 2010; Benjamini & Hochburg, 1995). Statistical significance for all tests was set at an alpha level of 0.05.

EOG traces were visually examined with custom, in-house software. Nystagmus was identified by trained experimenters (first, third and sixth authors) as characteristic jerk nystagmus (Serra & Leigh, 2002), that is, slow drift of at least 4°/s from a fixation point immediately followed by rapid corrective motion in the opposite direction. Traces that did not demonstrate clear nystagmus were rejected. VOR gain was defined as slow phase eye velocity/chair velocity. Practically, this was calculated as the slope (velocity) of the nystagmus slow phase divided by chair velocity; these measurements were taken at the approximate peak velocity in each hemicycle.

As described previously, spectral power recorded during the *Adaptation* period was split into epochs, thus allowing for examination of adaptation trends across cycles of the chair. This was quantified by fitting logistic regression curves for the first six cycles of each condition for each participant. Six cycles were chosen as Pettorossi et al. (2013) had indicated that the adaptive effect occurred within four cycles or fewer, and thus six cycles would encompass the period of maximal adaptation. The beta coefficients of these curves were fed into LMMs, as described above, to compare desynchronisation and resynchronisation trends between asymmetric and symmetric conditions in each region of interest and each frequency. To further examine the relationship between the change in EEG power and the subject's perceptual response, TPE values and beta coefficients were correlated using Pearson's correlation test. TPE values were normalised for the correlation test by dividing TPE by 360°; thus, normalised TPE

represented what proportion of a full revolution was perceived by the participant. Normalised EEG power in the first cycle was also analysed using LMMs to compare the initial power, relative to baseline, in each band per condition. Differences in steady-state spectral power between conditions were investigated for the *Adaptation* and *Post-Adaptation* periods, using the same statistical methods.

To test for differences in connectivity between conditions, we used network-based statistics (NBS; Zalesky et al., 2010). This method of testing identifies a contiguous network of edges (i.e. pairs of electrodes, or nodes) in which the sum strength of all connections is maximised against a null distribution built using non-parametric permutation (1000 iterations). Each edge is tested using an ANOVA of the fitted LMM and the family-wise error rate (FWER) *P*-value is calculated for the cluster of edges that individually fall below the threshold *P*-value, which was defined at 0.01. Average magnitude-squared coherence and average imaginary coherence were used as the measures of connectivity for our networks, which directly compared the asymmetric and symmetric conditions separately for each period (*Baseline*, *Adaptation* and *Post-Adaptation*). The *NBR* package was used to run this analysis (Gracia-Tabuenca & Alcauter, 2020). Networks were visualised using BrainNet Viewer (Xia et al., 2013).

## Results

In reporting their perceived head orientation, three participants gave a TPE greater than 3 SD away from the relevant average response. As a result, they were excluded from analysis of the psychophysical response ($n = 22$ subjects). Additionally, as the quality of EOG recordings was extremely variable both within and between participants, only clearly defined nystagmus was measured to calculate VOR gain, resulting in 11 participants with poor quality eye traces (probably due to roving eye artefacts) being excluded from analysis of the VOR response ($n = 14$ subjects). Although only a small pool was available for analysis, note that the purpose of measuring VOR gain was simply to confirm adaptation had not occurred; this response has already been thoroughly characterised in previous work (Faralli et al., 2022; Panichi, Faralli et al., 2017; Panichi, Occhigrossi et al., 2017; Pettorossi et al., 2013) and as such was not a main focus of this study.

### Psychophysical response

A repeated-measures ANOVA of the fitted LMM revealed that participants reported greater TPE during asymmetric rotations [$F_{(1, 63)} = 24.05$, $P < 0.0001$] (Fig. 2*A*). In essence, participants' subjective direction of facing

drifted considerably in the direction of the FHC during asymmetric oscillations. Neither frequency of stimulation nor the interaction between frequency and symmetry were significant [$F_{(1, 63)} = 3.16$ and 0.02, $P = 0.081$ and 0.887, respectively].

All incorrect perceived facings reported during the asymmetric conditions [LF: mean (SD) = $-194.32°$ (233.78); HF: mean (SD) = $-139.09°$ (163.09)] were to the left of the reference point (in the direction of the FHC), indicating a perceptual 'neglect' of the SHC. There was variability in the magnitude of adaptation; in 59% of the participants' adaptation TPE was at least 45°. Approximately 73% of participants also experienced perceptual error in response to LF symmetric stimulation [LF: mean (SD) = $-49.09°$ (185.75); HF: mean (SD) = $-2.05°$ (94.68)] but, critically, perceptual error was reported in both directions, with a slight bias

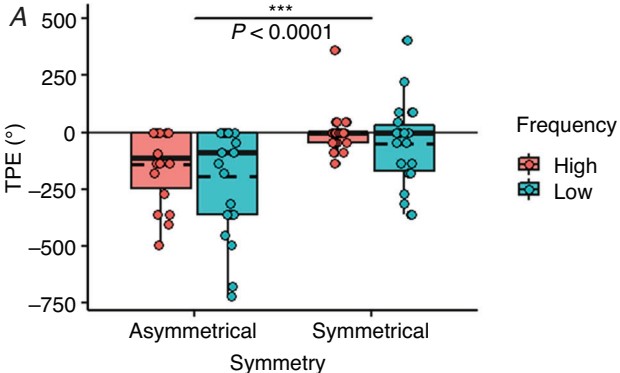

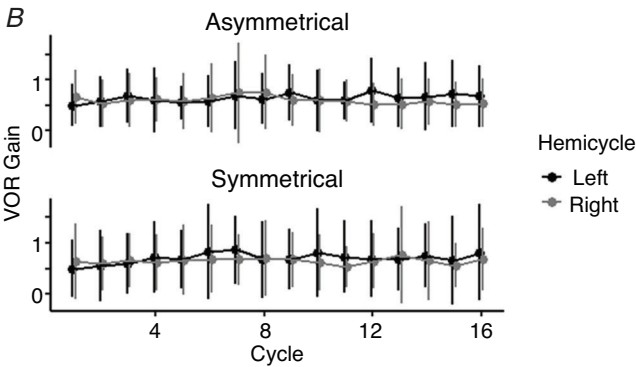

**Figure 2. Psychophysical and VOR response to symmetric and asymmetric stimulation**
*A*, boxplots depicting total position error (TPE) in degrees for each condition with raw data points overlaid ($n = 22$ subjects, 88 observations). The solid and dashed lines in each box represent the median and mean TPE, respectively. The boxes extend from the first to the third quartile of the data; whiskers extend to a maximum of 1.5 times the interquartile range. Negative values indicate leftward error. The *P* value indicates significant comparison between the two asymmetric and two symmetric conditions. *B*, line graph depicting average VOR gain for 14 subjects ($n = 1120$ observations) across the *Adaptation* period. Hemicycles correspond to leftward (black) or rightward (grey) movement of the chair. Error bars indicate ± 2 SD.

towards the left (indicated by leftward error relative to the reference landmark). However, the average magnitude of this error was still much smaller than that reported in the asymmetric conditions (Fig. 2*A*).

### VOR response

Given the smaller sample size for analysis of VOR gain, and the lack of significance for stimulus frequency when analysing TPE, VOR gain was averaged across the two asymmetric and the two symmetric conditions for statistical analysis. Average gain was then calculated for each hemicycle (right and left) of each cycle across all participants. Visual inspection of these measurements showed that VOR gain in each hemicycle did not change over the course of the 16 cycles, for either the asymmetric or the symmetric conditions (Fig. 2*B*). To confirm this, an LMM was fit for the difference of VOR gains between the first two and last two cycles (Last – First), examining the effects of symmetry and hemicycle [Asymmetric-Left: mean (SD) $= 0.08$ (0.08); Asymmetric-Right: mean (SD) $= -0.05$ (0.12); Symmetric-Left: mean (SD) $= 0.13$ (0.30); Symmetric-Right: mean (SD) $= 0.03$ (0.22)]. The repeated-measures ANOVA run on the LMM did not show any significant changes in gain in response to symmetry or hemicycle [$F_{(1,28)} = 0.48$ and 2.51, $P = 0.502$ and 0.144, respectively]. The interaction between symmetry and hemicycle was also not significant [$F_{(1,28)} = 0.05$ and $P = 0.827$]. Thus, in line with previous findings (Faralli et al., 2022; Panichi, Faralli et al., 2017; Panichi, Occhigrossi et al., 2017; Pettorossi et al., 2013), asymmetric oscillation only induced adaptation in the perceptual but not the reflexive ocular response.

### Time-based spectral analysis

To simplify further analysis, statistical examination of all EEG measures was conducted on the LF asymmetric and symmetric conditions only. We chose to focus on these conditions given the results of the psychophysical analysis – recall that the frequency of stimulation was not found to be a significant factor – and because the greater degree of asymmetry present in the asymmetric slow condition would probably emphasise any difference in the EEG response to asymmetric *versus* symmetric stimulation.

Spectrograms ($n = 24$ subjects) were analysed to investigate temporal changes in EEG power during the *Adaptation* period. Close visual examination of spectrograms at various electrodes was utilised to roughly identify trends in the data. This preliminary analysis yielded several important observations. First, the alpha frequency band had the most modulation over the course of the *Adaptation* period. This was most apparent in the

first few cycles of adaptation; both symmetry conditions showed a desynchronisation of alpha oscillations relative to baseline, although this was more pronounced in the symmetric condition. This can be clearly observed in the occipital region, which is known to receive substantial propagation of alpha rhythms from subcortical generators (Halgren et al., 2019); the O2 electrode, depicted in Fig. 3*A*, demonstrates this well. Alpha rhythms then began to resynchronise toward baseline values within the next few cycles. This pattern was observed in several regions, but most prominently in occipital and parietal regions, as well as in frontal and prefrontal electrodes. The other observation of interest was that the EEG response did not appear to differ between the FHC and SHC of any given cycle.

Regions of interest were partially selected *a priori*, based on previous work postulating the role of the parieto-occipital cortices in vestibular processing (Gale et al., 2016; Lopez & Blanke, 2011) as well as the role of frontoparietal attentional networks in error processing and sensory adaptation (Corbetta & Shulman, 2002; Petersen & Posner, 2012; Scolari et al., 2015). The decision to focus on these regions was further confirmed through visual examination of the average normalised power of each spectrogram (Fig. 3*B*), which aligned with the previously noted observations; there was a greater suppression of alpha power in the symmetric condition relative to the asymmetric condition and this was largely reflected in the parietal (P7, P3, Pz, P4 and P8 electrodes), occipital (O1, Oz and O2 electrodes), prefrontal (Fp1, Fpz and Fp2 electrodes) and frontal (F7, F3, Fz, F4 and F8 electrodes) regions (Fig. 3*B*). There was no observable lateralisation of this response. As a result, power was averaged across the electrodes in these four regions, henceforth referred to as regions of interest (ROIs), for further analysis (see Fig. 4*A* for a topographic map of these ROIs). IAF – the point at which power was measured – averaged (SD) at 10.47 Hz (0.85 Hz) across all regions, with values ranging from 8.43 to 12.68 Hz.

Adaptation over the first six cycles ($n = 4202$ data points) was characterised by a logistic regression curve; a repeated-measures ANOVA of the beta coefficients revealed a significant interaction between region and condition for alpha frequency only [$F_{(3, 157)} = 3.15$, $P = 0.0153$]. *Post hoc* testing revealed that adaptation curves diverged in the occipital and prefrontal regions, for the asymmetric condition only ($t = 2.85, P = 0.0482$). Alpha power in the asymmetric condition recovered very quickly in prefrontal areas, but barely at all in the occipital region, while the desynchronisation and rebound response between the two regions was similar for the symmetric condition (Fig. 4*B*). The correlation between normalised TPE and the beta coefficients was not significant in either the occipital or the prefrontal region for the asymmetric (occipital: $R = -0.067$, $P = 0.768$;

prefrontal: $R = 0.079$, $P = 0.727$) or the symmetric condition (occipital: $R = -0.130$, $P = 0.574$; prefrontal: $R = -0.035$, $P = 0.880$).

ANOVAs of normalised power in the first cycle in each condition and ROI ($n = 940$ data points) revealed a significant main effect of condition for alpha [$F(1, 202) = 4.60$, $P = 0.0332$], beta [$F(1, 202) = 5.15$, $P = 0.0244$] and theta [$F(1, 202) = 4.77$, $P = 0.0301$] frequencies; the main effect of condition was insignificant for delta frequency [$F(1, 202) = 1.16$, $P = 0.283$] (Fig. 4C). The effect of region was significant for beta frequency [$F(4, 202) = 5.15$, $P = 0.024$] but not the other three frequency bands [alpha: $F(4, 202) = 0.83$, $P = 0.507$; delta: $F(4, 202) = 2.17$, $P = 0.0737$; theta: $F(4, 202) = 1.43$,

$P = 0.225$]. The interaction between condition and region was not significant for any frequency band [alpha: $F(4, 202) = 1.70$, $P = 0.151$; beta: $F(4, 202) = 1.52$, $P = 0.197$; delta: $F(4, 202) = 0.58$, $P = 0.681$; theta: $F(4, 202) = 1.67$, $lP = 0.159$].

## Steady-state spectral power

To compare spectral power both between conditions and between the *Adaptation* and *Post-Adaptation* periods, as well as further investigate EEG spatial patterns of the adaptation and 'neglect' experienced by subjects, an analysis of steady-state spectral power was conducted. As in the previous section, visual examination was conducted

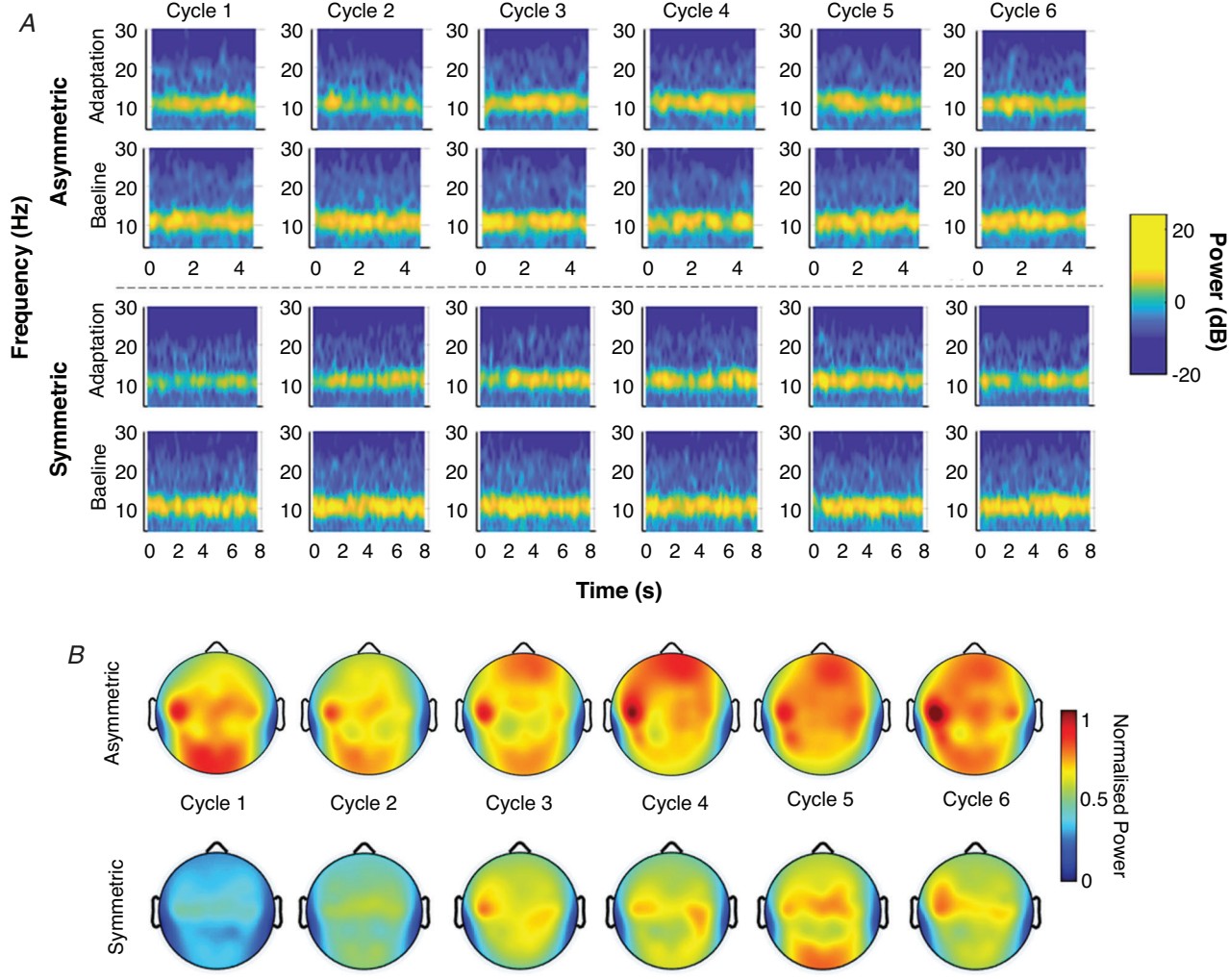

**Figure 3. Spectral power changes associated with repeated symmetric and asymmetric oscillations**
*A*, spectrograms for the first six cycles of the low-frequency asymmetric and symmetric conditions. Plots depict average spectral power for all subjects ($n = 25$) in the O2 electrode. Power was strongest in the alpha frequency band (yellow band) particularly during baseline. Alpha power was reduced during adaptation, more so for the symmetric condition than the asymmetric condition. *B*, normalised alpha power for the first six cycles in 30 electrodes ($n = 25$ subjects) in the low-frequency asymmetric and symmetric conditions. Dark red indicates that power during the *Adaptation* period is equal to or greater than power during baseline (synchronisation); dark blue indicates power in the *Adaptation* period is greatly reduced compared to baseline (desynchronisation).

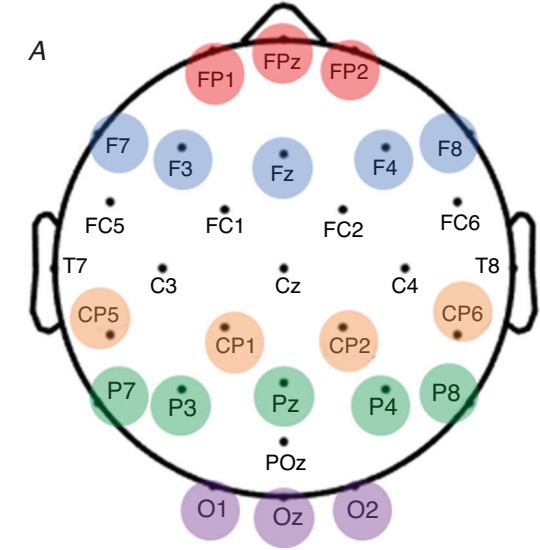

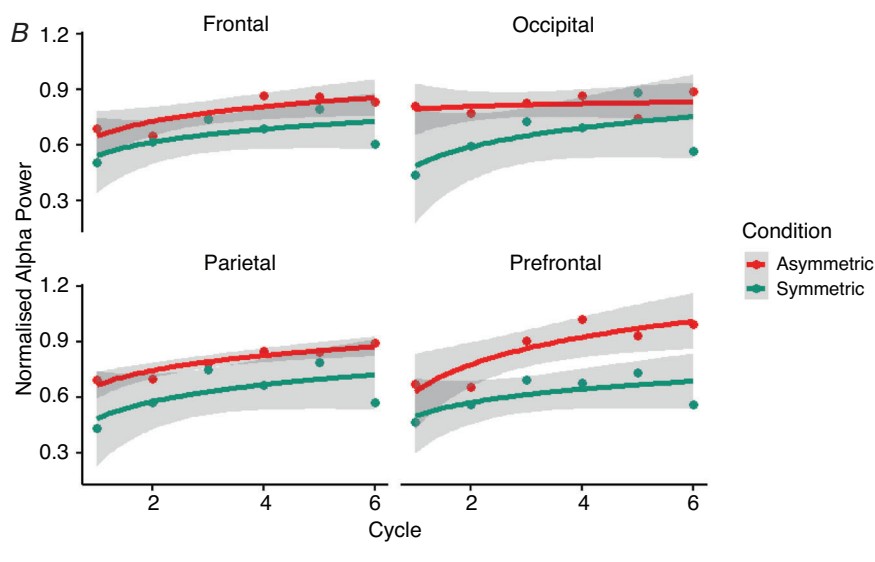

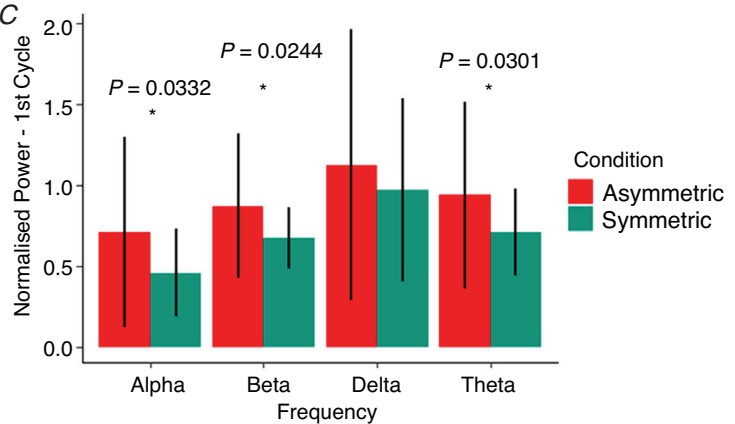

**Figure 4. Regional analysis of changes to spectral power during symmetric and asymmetric**

*A*, blank topographic map showing regions of interest: centroparietal (orange), frontal (blue), occipital (purple), parietal (green), prefrontal (red). *B*, average normalised alpha power for the first six cycles (*n* = 4202 data points) of the *Adaptation* period for four regions of interest (ROIs). The trend line is the average log-fit curve for each condition in that ROI. Grey shading indicates the 95% confidence interval. *C*, average normalised power (*n* = 940 data points) for the first cycle in the four frequency bands (averaged across all ROIs). Error bars represent ± 1 SD.

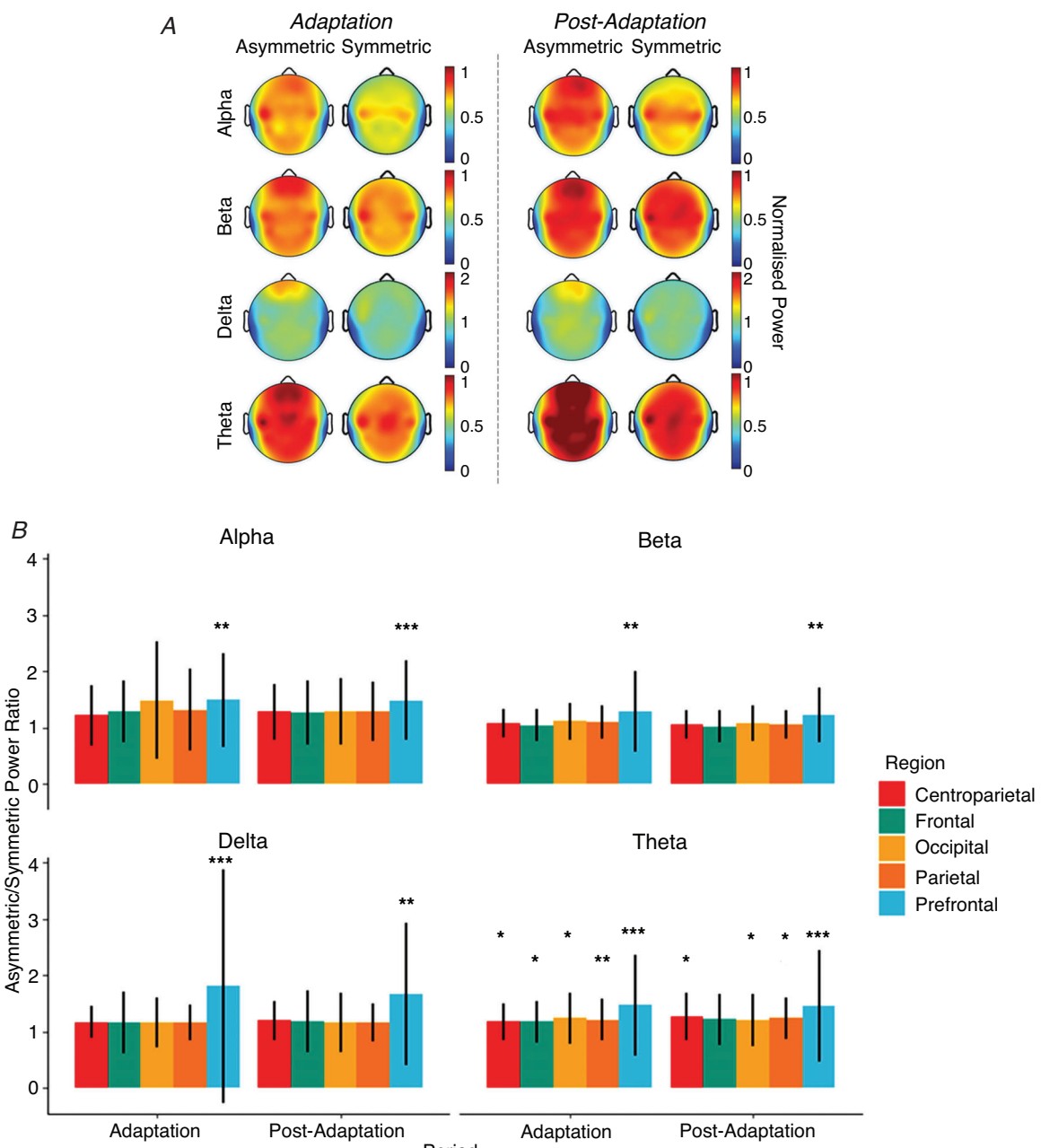

**Figure 5. Results of steady state spectral analysis across region, condition, and frequency**
*A*, average steady power during and after adaptation for the symmetric and asymmetric condition in each frequency band for 30 electrodes. Power is normalised by baseline values. Dark red indicates that power is equal to or greater than baseline values (synchronisation); dark blue indicates power is greatly reduced compared to baseline (desynchronisation). *B*, average ratio of asymmetric/symmetric normalised power values in five ROIs for each frequency band during and after adaptation. Error bars indicate ± 1 SD. Values greater than 1 indicate that power in the asymmetric condition was greater than power in the symmetric condition. Significant pairwise comparisons between the two conditions for a given ROI are indicated with asterisks (see Table 1 for exact values). *$P = 0.01-0.05$; **$P = 0.0001-0.01$; ***$P < 0.0001$.

on normalised steady-state power head plots to determine ROIs to analyse. No lateralisation was observed in either the *Adaptation* or the *Post-Adaptation* period in any frequency band, but several regions (most of which were predicted to be relevant *a priori*) appeared to differ

between conditions, including frontal, prefrontal, parietal, centroparietal and occipital electrodes (Fig. 5*A*). These electrodes were grouped by region (same ROIs as before, but with the centroparietal electrodes – CP5, CP1, CP2 and CP6 – included; see Fig. 4*A*) and averaged for further

analysis; these values were then used to construct the LMMs ($n = 7319$ data points).

There was a significant interaction between ROI and condition for both the *Adaptation* and the *Post-Adaptation* periods in the alpha [$F (4, 887) = 3.50$ and $F (4, 891) = 6.91$, $P = 0.008$ and $< 0.0001$, respectively], beta [$F (4, 882) = 5.66$ and $F (4, 874) = 3.84$, $P = 0.0002$ and $0.004$], delta [$F (4, 878) = 5.88$ and $F (4, 870) = 5.42$, $P = 0.0001$ and $0.0003$] and theta [$F (4, 894) = 3.80$ and $F (4, 879) = 3.67$, $P = 0.005$ and $0.006$] frequency bands (Fig. 5*B*). *Post hoc* analysis revealed that, in all cases, this interaction was driven by a significant difference in normalised power between conditions in the pre-frontal region, apart from the theta band which differed significantly across all ROIs (see Table 1).

### Network-based statistic

To examine connectivity of cortical networks during each condition, a NBS was calculated for each frequency band for the *Adaptation* and *Post-Adaptation* periods using coherence values. LMMs were used when comparing networks between conditions. The only significantly different networks between conditions were for the beta (*FWER-P* = 0.004) and delta frequency bands (*FWER-P* = 0.007) in the *Adaptation* period; these networks constituted 14 and 17 connections, respectively (Fig. 6*A*).

These findings suggest that, while experiencing asymmetric stimulation, connections between frontal and parieto-occipital regions are increasingly coherent for both delta and beta frequencies. Closer visual examination of these networks (Fig. 6*A*) indicates that, in the beta frequency, there is considerable involvement of frontal, central and parietal regions bilaterally, and the left occipital lobe. The network identified in delta, however, was observed to be lateralised to the right, with multiple frontoparietal and fronto-occipital connections.

Concern around the results being attributable to volume conduction (VC) was minimal, given that we utilised a within-subjects design (i.e. many factors influencing VC were inherently controlled for), and given the distance between nodes (electrodes) comprising edges (connections). However, there was a cluster of neighbouring electrodes identified in the beta network

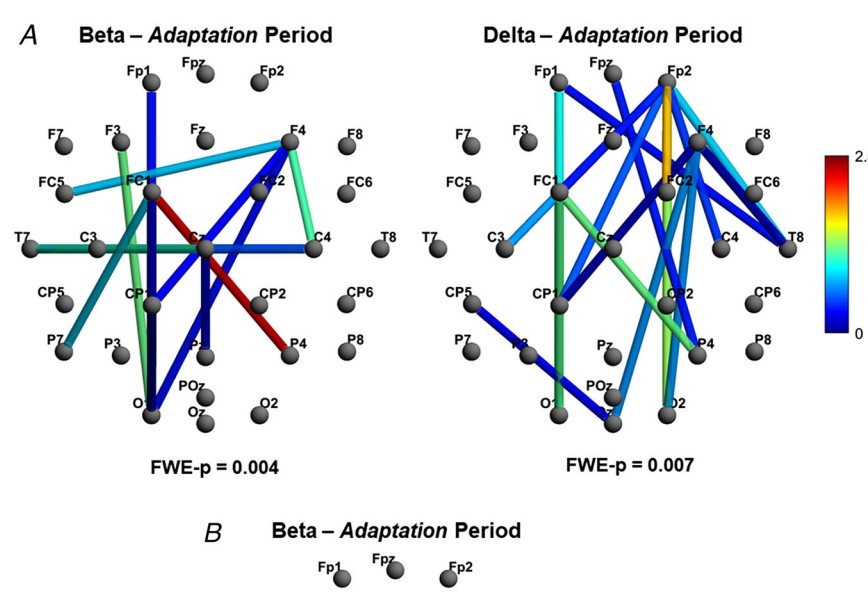

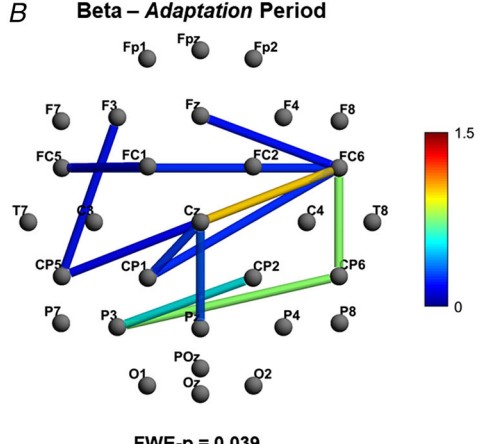

**Figure 6. Functional connectivity networks associated with perceptual adaptation**

*A*, significant networks in beta and delta frequencies during adaptation identified using the network-based statistics (NBS) and using magnitude-squared coherence. *B*, significant network identified in the beta frequency band during adaptation using absolute imaginary coherence. Networks were visualised with BrainNet Viewer. The key (colour bar) to the right represents a measure of the positive difference in the strength of the connection between two nodes in the asymmetric *versus* symmetric conditions. Note that 'strength' is a unitless measure calculated as part of the NBS. Dark red indicates greater difference in strength between asymmetric and symmetric conditions (asymmetric > symmetric) and deep blue indicates minimal but still significant difference between conditions.

**Table 1. *Post hoc* test results for normalised EEG steady-state spectral analysis**

| Frequency | Period | Region | Mean (SD) | | t-statistic | P |
|---|---|---|---|---|---|---|
| | | | Asymmetric | Symmetric | | |
| Alpha | *Adaptation* | CP | 0.73 (0.32) | 0.65 (0.23) | 1.24 | 0.216 |
| | | F | 0.76 (0.29) | 0.64 (0.22) | 1.56 | 0.120 |
| | | O | 0.76 (0.40) | 0.63 (0.29) | 1.88 | 0.0612 |
| | | P | 0.71 (0.30) | 0.61 (0.25) | 1.48 | 0.137 |
| | | PF | 0.81 (0.48) | 0.61 (0.23) | 2.91 | **0.0037** |
| | *Post-Adaptation* | CP | 0.89 (0.30) | 0.78 (0.31) | 1.70 | 0.0890 |
| | | F | 0.84 (0.27) | 0.75 (0.31) | 1.46 | 0.144 |
| | | O | 0.77 (0.30) | 0.74 (0.41) | 0.45 | 0.656 |
| | | P | 0.81 (0.28) | 0.72 (0.32) | 1.59 | 0.116 |
| | | PF | 0.98 (0.73) | 0.71 (0.30) | 4.09 | **<0.0001** |
| Beta | *Adaptation* | CP | 0.79 (0.21) | 0.75 (0.15) | 1.02 | 0.310 |
| | | F | 0.76 (0.21) | 0.76 (0.19) | 0.13 | 0.901 |
| | | O | 0.79 (0.18) | 0.75 (0.16) | 1.01 | 0.312 |
| | | P | 0.78 (0.19) | 0.75 (0.15) | 1.13 | 0.259 |
| | | PF | 0.90 (0.44) | 0.76 (0.20) | 3.18 | **0.0015** |
| | *Post-Adaptation* | CP | 0.93 (0.19) | 0.93 (0.22) | 0.40 | 0.689 |
| | | F | 0.90 (0.20) | 0.92 (0.25) | −0.33 | 0.741 |
| | | O | 0.86 (0.18) | 0.85 (0.24) | 0.49 | 0.622 |
| | | P | 0.88 (0.18) | 0.87 (0.23) | 0.61 | 0.544 |
| | | PF | 1.02 (0.35) | 0.90 (0.23) | 3.07 | **0.0022** |
| Delta | *Adaptation* | CP | 1.03 (0.24) | 0.92 (0.17) | 1.26 | 0.209 |
| | | F | 0.97 (0.32) | 0.95 (0.23) | 0.19 | 0.850 |
| | | O | 1.05 (0.28) | 0.96 (0.25) | 0.98 | 0.330 |
| | | P | 1.05 (0.26) | 0.94 (0.21) | 1.44 | 0.149 |
| | | PF | 1.35 (1.10) | 1.00 (0.50) | 4.16 | **<0.0001** |
| | *Post-Adaptation* | CP | 1.11 (0.30) | 0.98 (0.22) | 1.37 | 0.172 |
| | | F | 1.04 (0.40) | 0.98 (0.32) | 0.63 | 0.529 |
| | | O | 1.08 (0.39) | 1.02 (0.33) | 0.67 | 0.499 |
| | | P | 1.09 (0.31) | 0.99 (0.44) | 1.26 | 0.207 |
| | | PF | 1.39 (1.22) | 1.01 (0.45) | 4.07 | **0.0001** |
| Theta | *Adaptation* | CP | 0.92 (0.32) | 0.80 (0.21) | 2.56 | **0.0106** |
| | | F | 0.89 (0.32) | 0.80 (0.25) | 2.16 | **0.0308** |
| | | O | 0.90 (0.25) | 0.79 (0.21) | 2.52 | **0.0120** |
| | | P | 0.89 (0.28) | 0.78 (0.20) | 2.63 | **0.0087** |
| | | PF | 1.02 (0.49) | 0.78 (0.24) | 5.01 | **<0.0001** |
| | *Post-Adaptation* | CP | 1.11 (0.38) | 0.96 (0.30) | 2.55 | **0.0108** |
| | | F | 1.03 (0.33) | 0.94 (0.31) | 1.65 | 0.0993 |
| | | O | 1.13 (0.38) | 1.00 (0.32) | 2.12 | **0.0339** |
| | | P | 1.09 (0.35) | 0.95 (0.30) | 2.40 | **0.0164** |
| | | PF | 1.23 (0.80) | 0.94 (0.29) | 4.37 | **<0.0001** |

Results of pairwise comparison, using *emmeans*, for steady-state data (*n* = 7319 data points). Significant *P* values are in bold type. CP, centroparietal; F, frontal; O, occipital; P, parietal; PF, prefrontal.

(CP1, CPz and Pz) that were identified as potentially susceptible to VC. To further investigate this, the data were reprocessed using imaginary coherence (Nolte et al., 2004). This analysis identified a network of 12 connections in the beta band only (*FWER-P* = 0.039), primarily between bilateral frontocentral and centroparietal electrodes (Fig. 6*B*).

## Discussion

The present study aimed to investigate EEG correlates of the acute perceptual neglect induced by asymmetric vestibular stimulation. Our results replicated previous findings that asymmetric stimulation can induce perceptual neglect of the SHC and that this is not

reflected in brainstem reflex mechanisms such as VOR (Faralli et al., 2022; Panichi, Faralli et al., 2017; Panichi, Occhigrossi et al., 2017; Pettorossi et al., 2013). As these psychophysical and reflexive responses have already been thoroughly described in the aforementioned studies – and given that the main aim of the present study was to determine the EEG markers of the perceptual adaptation – we will briefly comment on the behavioural importance of dissociating the two responses but will not discuss our findings around these measures further.

First, it was noted that there was a considerable degree of error in the low-frequency symmetric condition, although this error was noted in both directions. We attribute this to the stimulus frequency; direction detection thresholds rise below 0.2 Hz rotations (Carriot et al., 2014; Grabherr et al., 2008; Mallery et al., 2014), and as such some participants may have perceived all motion in one direction. This is still behaviourally distinct, however, from the considerable adaptation noted with the asymmetric stimulus, the proposed purpose of which is discussed in greater detail below.

It is also worth noting that, in this paradigm, conflict arises at multiple levels. Slow *versus* fast rotations in opposing directions is one such conflict (or contrast), but not necessarily the only important one. The thalamo-cortical mechanisms responsible for perception and the brainstem circuits that mediate VOR unquestionably dissociate, giving rise to some form of internal error in which perception is altered but VOR is not. Dissociation of reflexive and perceptual responses has been previously documented as an adaptive mechanism in ballet dancers, who are exposed to repetitive vestibular stimulation, such as pirouettes, as well as in vestibular patients (Cousins et al., 2013; Nigmatullina et al., 2015; Seemungal, 2014). The uncoupling of these responses allows for more behavioural flexibility, especially when one response is made unreliable, as is the case here. That said, the precise mechanisms allowing for this uncoupling are still not well understood and further investigation is required to elucidate the exact nature of its role in this paradigm.

### Role of alpha frequency band in adaptive vestibular perception

One of the main findings of this study was that power in the alpha frequency band is associated with the adaptive process induced by asymmetric rotation. Specifically, desynchronisation and subsequent rebound of alpha power differs during asymmetric *versus* symmetric stimulation, particularly in the prefrontal and occipital regions, where the resynchronisation is accelerated and diminished, respectively. Body motion has been shown to correspond with a reduction of alpha power in central

parietal and lateral parieto-temporal areas (Gutteling & Medendorp, 2016). Gale et al. (2016) additionally showed that a continuously perceived vestibular stimulation is associated with suppression of alpha over the parietal cortex. Our findings show the same initial reduction, which then decreases over the course of continual stimulation. This trend has been noted in the literature previously; a relative increase after initial reduction was noted by Gale et al. (2016), as well as in separate studies using vection to induce motion perception (Harquel et al., 2020; McAssey et al., 2020).

Our findings, along with previous work, suggest that the initial alpha reduction indicates neural pathways are primed to detect and process incoming information, but then are fine-tuned to process salient information. This supposition, which is also discussed in similar work (Harquel et al., 2020; Pettorossi et al., 2013) is supported by work on the role of enhanced alpha as an inhibitory gating mechanism, one that facilitates attention by suppressing irrelevant information (Foxe & Snyder, 2011; Peng et al., 2015; Zani et al., 2020). In our paradigm, the asymmetric oscillations deliver conflicting stimulation in the form of fast *versus* slow rotation in opposite directions. As the FHC is a more salient stimulus (recall the vestibular system is a high-pass filter), the increased resynchronisation in the prefrontal region could indicate a suppression of less salient information, namely the SHC. This explanation agrees with the literature on the role of the prefrontal cortex in monitoring conflict and error and providing feedback to posterior regions to adjust future behaviour and increase feed-forward of relevant stimuli (Buschman & Miller, 2007; Zavala et al., 2018).

The finding that alpha suppression did not recover at the same rate in occipital regions during the asymmetric condition was unexpected. Given the theorised role of the parietal cortex in vestibular processing (Cullen, 2019; Lopez & Blanke, 2011; Lopez et al., 2012), and previous findings that alpha suppression during vestibular stimulation was strongest over the parietal lobe (Gale et al., 2016; Gutteling & Medendorp, 2016), it was hypothesised that the greatest difference in the pattern of alpha desynchronisation and rebound between the asymmetric and symmetric conditions would occur here. One possible explanation is that the activation of the occipital cortex reflects mental imagery or visualisation strategies to maintain orientation with their initial reference point (in the present study, the air conditioner). Landmark-based orienting and navigation has been shown to recruit regions in the occipital–temporal and retrosplenial cortices (Committeri et al., 2004); retrieval of spatial information encoded using an allocentric frame of reference, such as landmark-based navigation, is correlated with alpha suppression in occipital regions (Chiu et al., 2012). The suppression of occipital alpha power noted in the present study could indicate that

participants were utilising this landmark-based strategy; this is probable, given our experimental paradigm. Thus, as adaptation in the asymmetric condition alters perception of their facing, persistent suppression of alpha could reflect the need for participants to continuously update their perception of the reference point location.

Although we were not able to directly correlate EEG desynchronisation and resynchronisation trends with perceptual error, it has been shown that adaptation to conflicting stimulation, such as the contrast between the opposing directionality and velocity of the FHC and SHC, can occur even without subjective awareness (Jiang et al., 2018). This would therefore suggest that the underlying physiological patterns are consistent regardless of subjective experience. As such, we feel reasonably confident that our EEG data, at a group level, still capture relevant cortical markers associated with perceptual adaptation, or 'neglect', of vestibular stimulation. Future work may elucidate a direct relationship between the perceptual and physiological measures of this adaptation.

### Frontoparietal attentional networks mediate vestibular perceptual 'neglect'

Increased connectivity during and after adaptation was primarily noted between frontal and parietal regions for the beta and delta frequency bands for the asymmetric condition. Considering the suggestion above, that participants reorient to attend to the more salient stimuli, we propose that the increased connectivity is reflective of greater attentional resources being allocated to complete the task at hand. Participants know, regardless of the stimulation they experience, they must report their perceived facing at the end, a clear goal that would be facilitated by the top-down mechanisms of the dorsal attentional network (Corbetta & Shulman, 2002; Petersen & Posner, 2012). When the stimulation is conflicting (recall that the differing velocity and directionality of the FHC and SHC can be considered conflicting stimuli), the bottom-up reorienting mechanisms of the ventral attentional network are recruited to ensure the end goal is still achieved. This may be facilitated by increased delta–beta frequency coupling in parietal regions, which has been associated with increased attentional control and orienting attention (Morillas-Romero et al., 2015).

The results of the imaginary coherence analysis were unexpected, as the network identified comprised almost exclusively fronto-central and centroparietal connections. However, it has been theorised that midline regions can reflect late-latency vestibular signals (Nakul et al., 2021; Todd et al., 2014), which may reflect some of the bottom-up feed-forward mechanisms discussed previously.

Delta frequency, which is believed to be generated in frontal regions, is also implicated in promoting connectivity of the frontoparietal control network; enhanced delta power is indicative of suppression of irrelevant information when increased concentration is needed (Alper et al., 2006; Harmony, 2013). This aligns with two main findings. First, significantly enhanced delta power was observed during the asymmetric condition, both during and after adaption, in the prefrontal regions. Second, a clear frontoparietal network was identified in our analysis of delta coherence, and this network is more coherent during asymmetric stimulation. Both these findings suggest that delta mediates the perceptual 'neglect' resultant from asymmetric rotation, and align with the assertion, given in the section above, that alpha inhibits extraneous information to facilitate better task performance.

In addition, it has been shown that physiological impairment of the ventral attentional network, which mediates non-spatial functions such as attentional reorienting and arousal, decreases feed-forward interaction with the ipsilateral dorsal network, resulting in spatial neglect of left hemispace (Corbetta & Shulman, 2011). It is possible that our paradigm mimics this impairment, where the stimulus drives a reduced alerting of less salient stimuli (i.e. the SHC), which ipsilateral dorsal networks responsible for spatial orienting then ignore. However, given that this perception is driven by the stimulus characteristics, not neurological damage, further work would need to be undertaken to properly investigate this view.

### Potential role of subcortical circuits

An obvious limitation of this study is that EEG is only capable of capturing superficial cortical activity. Consequently, we are not able to provide evidence of the role of subcortical or thalamic regions in the adaptive process in this study, but we can look to existing literature to hypothesise the potential role of these structures.

As stated earlier, when vestibular perception is disrupted, as in the asymmetric condition, alpha recovery may indicate that there is no incoming information to process. This could suggest that the vestibular signal is being interrupted before its arrival in the cortex and, therefore, the adaptive response is happening in subcortical circuits, such as the ascending head direction (HD) system. HD cells encode directional heading in the yaw plane and discharge when an animal moves into a specific allocentric head orientation (Taube, 2007). Given that in the present study subjects are passively rotated in the yaw plane and must report their perceived facing by referencing external landmarks, the HD system must be involved to some degree.

The HD pathway receives signals from vestibular nuclei in the brainstem which are passed first to the dorsal tegmental nucleus, then to the lateral mammillary nucleus (LMN) in the hypothalamus, the anterodorsal thalamic nucleus and then the post-subiculum in the cortex; signals are then passed to the hippocampus via the retrosplenial and entorhinal cortices with some reciprocal connections to the LMN (Peyrache et al., 2019; Taube, 2007; Yoder & Taube, 2014). HD cells, which play an important role in our ability to navigate, are influenced by both body-internal and body-external information, referred to as idiothetic and allothetic cues, respectively (Taube, 2007). The paradigm presented here requires a type of navigation known as path integration, which refers to the ability to update directional orientation using only idiothetic cues – in our case, vestibular information. In the present study, our subjects know their initial orientation but must update their internal cognitive map using only perceived vestibular cues.

Interestingly, the preferred firing direction (PFD) of HD cells can be altered during path integration, a form of error monitoring that allows for improved performance on subsequent navigation tasks. Valerio & Taube (2012) propose two mechanisms by which this takes place: resetting, in which HD cells are re-tuned to their original PFD, or remapping, in which the cell's PFD is set to a new orientation for all subsequent trials using available landmarks. Remapping is more likely to occur in the context of large, disorienting errors or multiple consecutive smooth turns (Valerio & Taube, 2012) – such as those experienced during slow asymmetric rotation. Hence, it is possible that in the context of only one type of idiothetic cue, subjects form an allocentric frame of reference based on remembered landmarks but, as stimuli become unreliable, subjects must continuously remap their reference point between rotations to maintain orientation. Further work would need to be undertaken to validate this hypothesis, specifically investigating the role of the HD system during this type of vestibular adaptation.

A separate consideration is the possible influence of low-level proprioceptive signals on the adaptive response. Although every measure was taken to restrict proprioceptive input during the experimental set-up, it is possible there was a proprioceptive contribution to the perceptual response. However, it is unlikely that this had a significant effect on the adaptation noted. Increased stimulus frequency has been shown to increase the gain of the vestibulo-perceptual response, but not the proprioceptive response (Mergner et al., 2001; Nakamura & Bronstein, 1995), meaning the increased acceleration of the FHC would be sensed less efficiently by the proprioceptive system relative to the vestibular system. Further evidence of this is provided by previous work which found that passive head deviations during asymmetric rotations

had no impact on movement perception (Panichi et al., 2011). Additionally, studies of this paradigm in vestibular neuritis (Panichi, Faralli et al., 2017) and Meniere's disease (Faralli et al., 2022) populations have shown that vestibular impairment alters the adaptive response noted in healthy individuals. Together, these findings, along with those in the present study, support that the adaptation is due to altered vestibular perception.

In summary, asymmetric whole-body oscillation induces acute perceptual 'neglect' of rotation in the slower direction without moderation of peripheral and low-level vestibular mechanisms, such as VOR. Our data suggest this altered perception is mediated by alpha inhibitory mechanisms in the prefrontal cortex and by increased connectivity of frontoparietal attentional networks, which may exert top-down control over subcortical circuits involved in spatial orientation. Overall, this paradigm may serve as a model for providing additional insight into the cortical processes underpinning spatial neglect and vestibular adaptation.

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

## Additional information

### Data availability statement

All data supporting the results have been included in the text and figures.

### Competing interests

The authors declare that they have no competing financial or personal interests that could have appeared to influence the work reported in this paper.

### Author contributions

J.I.C., P.C.A., V.E.P. and A.M.B. conceptualised and designed the research; J.I.C., O.G. and P.C.A. recruited subjects, performed the research and collected data; J.I.C., O.G., P.C.A., R.T.I. and A.M.B. analysed the data; J.I.C. wrote and prepared the manuscript; O.G., P.C.A., R.T.I., V.E.P. and A.M.B. reviewed and edited the manuscript. All authors approved the final version of the manuscript and agree to be accountable for the work in its entirety, as well as investigate and resolve any queries relating to the accuracy or integrity of the work. All persons listed as authors qualify for authorship, and all who qualify for authorship are listed.

### Funding

J.I.C. was supported by the Marshall Aid Commemoration Commission. The work was made possible by a research grant (R481/0516) from the Dunhill Medical Trust (to A.M.B.) and a block grant from the National Institute for Health Research Imperial Biomedical Research Centre. R.T.I. was supported by the Dunhill Medical Trust.

### Acknowledgements

We thank Mr David Buckwell for his technical support and many years of dedicated service to the group; they wish him well in retirement. We also thank Dr Qadeer Arshad for his input in the initial design of this work.

### Keywords

adaptation, alpha frequency, asymmetric rotation, EEG, frontoparietal, vestibular perception

## Supporting information

Additional supporting information can be found online in the Supporting Information section at the end of the HTML view of the article. Supporting information files available:

**Statistical Summary Document**
**Peer Review History**
**Figure S1**
**Figure S2**
**Dataset**

