## [Peer Review History · The Journal of Physiology]

Electroencephalographic Response to Transient Adaptation of Vestibular Perception

Josephine I Cooke, Onur Guven, Patricia Castro Abarca, Richard T Ibitoye, Vito E Pettorossi, and Adolfo M. Bronstein
DOI: 10.1113/JP282470

Corresponding author(s): Adolfo Bronstein (a.bronstein@imperial.ac.uk)

The following individual(s) involved in review of this submission have agreed to reveal their identity: Yong Gu (Referee #1); Ken D O'Halloran (Referee #2)

Review Timeline:

Submission Date:	22-Nov-2021
Editorial Decision:	21-Feb-2022
Revision Received:	06-May-2022
Editorial Decision:	17-May-2022
Revision Received:	23-May-2022
Accepted:	06-Jun-2022

Senior Editor: Richard Carson

Reviewing Editor: Jing-Ning Zhu

Transaction Report:

Dear Professor Bronstein,

Re: JP-RP-2022-282470 "Electroencephalographic Response to Transient Adaptation of Vestibular Perception" by Josephine I Cooke, Onur Guven, Patricia Castro Abarca, Richard T Ibitoye, Vito E Pettorossi, and Adolfo M. Bronstein

Thank you for submitting your manuscript to The Journal of Physiology. It has been assessed by a Reviewing Editor and by 3 expert Referees and I am pleased to tell you that it is considered to be acceptable for publication following satisfactory revision.

The reports are copied at the end of this email. Please address all of the points and incorporate all requested revisions, or explain in your Response to Referees why a change has not been made.

NEW POLICY: In order to improve the transparency of its peer review process The Journal of Physiology publishes online as supporting information the peer review history of all articles accepted for publication. Readers will have access to decision letters, including all Editors' comments and referee reports, for each version of the manuscript and any author responses to peer review comments. Referees can decide whether or not they wish to be named on the peer review history document.

Authors are asked to use The Journal's premium BioRender (<https://biorender.com/>) account to create/redrawn their Abstract Figures. Information on how to access The Journal's premium BioRender account is here: <https://physoc.onlinelibrary.wiley.com/journal/14697793/biorender-access> and authors are expected to use this service. This will enable Authors to download high-resolution versions of their figures.

I hope you will find the comments helpful and have no difficulty returning your revisions within 4 weeks.

Your revised manuscript should be submitted online using the links in Author Tasks Link Not Available.

Any image files uploaded with the previous version are retained on the system. Please ensure you replace or remove all files that have been revised.

REVISION CHECKLIST:

- Article file, including any tables and figure legends, must be in an editable format (eg Word)
- Abstract figure file (see above)
- Statistical Summary Document
- Upload each figure as a separate high quality file
- Upload a full Response to Referees, including a response to any Senior and Reviewing Editor Comments;
- Upload a copy of the manuscript with the changes highlighted.

- A potential 'Cover Art' file for consideration as the Issue's cover image;
- Appropriate Supporting Information (Video, audio or data set https://jp.msubmit.net/cgi-bin/main.plex?form_type=display_requirements#supp).

To create your 'Response to Referees' copy all the reports, including any comments from the Senior and Reviewing Editors, into a Word, or similar, file and respond to each point in colour or CAPITALS and upload this when you submit your revision.

I look forward to receiving your revised submission.

If you have any queries please reply to this email and staff will be happy to assist.

Yours sincerely,

Richard Carson

REQUIRED ITEMS:

-Author photo and profile. First (or joint first) authors are asked to provide a short biography (no more than 100 words for one author or 150 words in total for joint first authors) and a portrait photograph. These should be uploaded and clearly labelled with the revised version of the manuscript. See Information for Authors for further details.

-Please upload separate high-quality figure files via the submission form.

-A Statistical Summary Document, summarising the statistics presented in the manuscript, is required upon revision. It must be on the Journal's template, which can be downloaded from the link in the Statistical Summary Document section here: https://jp.msubmit.net/cgi-bin/main.plex?form_type=display_requirements#statistics

-Papers must comply with the Statistics Policy https://jp.msubmit.net/cgi-bin/main.plex?form_type=display_requirements#statistics

In summary:

-If n {less than or equal to} 30, all data points must be plotted in the figure in a way that reveals their range and distribution. A bar graph with data points overlaid, a box and whisker plot or a violin plot (preferably with data points included) are acceptable formats.

-If $n > 30$, then the entire raw dataset must be made available either as supporting information, or hosted on a not-for-profit repository e.g. FigShare, with access details provided in the manuscript.

-' n ' clearly defined (e.g. x cells from y slices in z animals) in the Methods. Authors should be mindful of pseudoreplication.

-All relevant ' n ' values must be clearly stated in the main text, figures and tables, and the Statistical Summary Document (required upon revision)

-The most appropriate summary statistic (e.g. mean or median and standard deviation) must be used. Standard Error of the Mean (SEM) alone is not permitted.

-Exact p values must be stated. Authors must not use 'greater than' or 'less than'. Exact p values must be stated to three significant figures even when 'no statistical significance' is claimed.

-Statistics Summary Document completed appropriately upon revision

-Please include an Abstract Figure. The Abstract Figure is a piece of artwork designed to give readers an immediate understanding of the research and should summarise the main conclusions. If possible, the image should be easily 'readable' from left to right or top to bottom. It should show the physiological relevance of the manuscript so readers can assess the importance and content of its findings. Abstract Figures should not merely recapitulate other figures in the manuscript. Please try to keep the diagram as simple as possible and without superfluous information that may distract from the main conclusion(s). Abstract Figures must be provided by authors no later than the revised manuscript stage and should be uploaded as a separate file during online submission labelled as File Type 'Abstract Figure'. Please ensure that you include the figure legend in the main article file. All Abstract Figures should be created using BioRender. Authors should use The Journal's premium BioRender account to export high-resolution images. Details on how to use and access the premium account are included as part of this email.

EDITOR COMMENTS

Reviewing Editor:

Methods Details:

Please specify in methods when the subjects are required to report their orientations in the room, the rule for defining nystagmus in EOG data and EEG preprocessing and analysis, as well as relevant ethical information, according to the reviewers's suggestions.

Comments to the Author:

The manuscript reports an interesting EEG correlates of vestibular asymmetric perception in healthy humans. Please extensively revise the manuscript according to the reviewers' suggestions.

Senior Editor:

Comments for Authors to ensure the paper complies with the Statistics Policy:

Please extend the provision of raw data to the EEG-derived measures. At present, they are restricted to the behavioural data.

It is appreciated that raw data have been provided in the supplemental materials.

An indication of individual responses on the figures would, however, also be beneficial.

Comments to the Author:

It is noted that, "data were analysed using linear mixed-effects models (LMMs)". Please provide a more complete description of these models (rather than a generic reference to the type of model), that includes specification of the manner in which the data from individual participants were included in the analysis design. In reporting the outcomes of these analyses, the degrees of freedom associated with each F ratio should be reported in every instance. It is also desirable that an appropriate corresponding measure of effect size (preferably including confidence intervals) be provided in every instance.

It appears that VOR gain was calculated for only 14 participants, whereas the Time-Based Spectral Analysis was based on 25 participants. Please provide a justification for the use of distinct pools of participants. Please provide evidence to support the assumption that the pattern of outcomes obtained for the 14 participants in respect of the VOR response could be generalised to the larger pool of 25 participants. In this context, consider that the "not significant" effect of frequency would almost certainly have been "statistically significant" (i.e. with a conventional NHST alpha level of 0.05), if the sample size had been 25 rather than 14. As alluded to above, the provision of effect size estimates will provide the reader with a basis upon which to judge whether an effect was likely to have been present. Relatedly, what were the patterns of outcomes for the subset of 14 participants, in respect of the EEG- derived measures?

REFEREE COMMENTS

Referee #1:

In the current study, Cooke et al., studied EEG correlate of vestibular asymmetric perception illusion when conflict fast and slow sinusoidal oscillations were provided. In the behavior, subjects typically ignored the slow oscillation by reporting the fast motion phase, whereas VOR typically faithfully reflects both motion phases. The authors did not find correlates of EEG with the perceptual error, unfortunately, however, there is a trend that during conflict sinusoidal asymmetric oscillations, there is a desynchronisation in the frontal cortex, and a resynchronization in the occipital cortex, indicating the frontal-parietal network is inhibiting some conflict signals during the asymmetric oscillation conditions. The motivation of the study is nice. The study is carefully conducted. The results are clearly presented. I only have a few comments.

1. The method is not clear enough about when the subjects are required to report their orientations in the room. Is it always within one cycle, or it could be more than one cycle? The landmark is 45 degrees apart. It seems a bit coarse. One cannot be more fine steps?
2. In the symmetric condition, the errors could be very large for the slow motion condition. For example, as shown in Figure 2A, some subjects show error more than one cycle (>360 deg). In these cases, did the subjects really understand the task, or the errors really reflect their sensory error?
3. All the figures are not clear enough. They should have higher resolution and quality.

Referee #2:

Thank you for submitting your manuscript to The Journal of Physiology.

You must begin the Methods section with the sub-heading "Ethical approval". The relevant information required in this section is already provided by the authors in the opening paragraph.

Please refer to "participants" rather than "subjects".

It would be helpful to include a statement that there were no adverse outcomes and no lasting disorientation in the participants at the close of the study.

Referee #3:

In the present manuscript, the authors first replicated previous findings that asymmetric body stimulation (rotation along the z-axis) induces perceptual errors (so-called "vestibular neglect") although it does not impact brainstem reflex mechanisms such as the VOR. Secondly, the authors showed that the alpha frequency band was related to the adaptive process induced by asymmetric rotation. Although the current experiment is of interest and appears to be well conducted, I have some concerns that should be taken into account.

Majors

1) Looking at Figure 1, the traces of the samples reflect the fact that the chair is manually operated, as the profiles of these traces do not correspond to pure sinusoids, which is particularly visible with regard to the velocity. The fact that the chair is manually operated is understandable if one considers that the authors wish to limit the electrical noise in a Faraday cage. However, if one considers that the vestibular system (here the semicircular canals) is sensitive to accelerations, one can assume that the acceleration profiles are even further away from pure sinusoids. Considering that the traces reported in the figure are probably among the most presentable traces, can the authors reassure the reader about the quality of the stimulus considering that the experimental aim is to evaluate a perceptual dimension (vestibular neglect) that arises from an acceleration sensitive system.

2) The authors report a dissociation between the VOR, which is unaffected by stimulus asymmetry, which is also a direct measure of vestibular function, and perception, which is particularly sensitive to this asymmetry and is thought to reflect a high level of vestibular interpretation. It turns out that the rotation of the participants to the right and to the left stimulates of course the vestibular system, but does also stimulate the proprioceptive system (through pressures on the body surface that differ according to the speed and acceleration of the body in space). In this respect, one may wonder whether the dissociation between VOR and perception (consider as vestibular neglect) in response to stimulus asymmetry does not only vestibular signals involvement in reflex activities (VOR) but vestibular as well as proprioceptive signals for perception. This should definitely be discussed.

3) The definition of alpha rhythm 8-14 Hz is atypical, and does not fit to the recommended standard (please see table 1 of Babiloni et al. 2020 for a recent review), which is important for comparing studies. The upper bound is in most cases equal to 12 Hz, and should never be higher than 13 Hz. The authors should define the alpha bandwidth in respect of these international guidelines, or at least discuss the reasons of this discrepancy.

4) The Time-Based Spectral Analysis section begins with several visual observations and (what seems to be) arbitrary choices. First, the authors stated that "Average normalised power of each cycle was also examined and confirmed the previously noted observations, namely that there was a greater suppression of alpha power in the symmetric condition relative to the asymmetric condition and that the parietal, occipital, prefrontal, and frontal regions showed this response more clearly (Figure 3B). There was no observable lateralisation of this response." The authors should run statistical analyses in order to support these observations (i.e. sign. difference from baseline, lateralization), prior to the ROI creation ("As a result, power was averaged across the electrodes in these four regions, henceforth referred to as regions of interest (ROIs), for further analysis »). Second, the authors should justify the choice for plotting O2 for Fig 3A. Also, why not plot the last cycle in Fig 3A?

5) The same goes for the ROI definition of Steady state spectral power analysis: "Head plots of steady state power showed no lateralisation either the Adaptation or Post-Adaptation in any frequency band, but several regions appeared to differ between conditions, including frontal, prefrontal, parietal, centroparietal, and occipital electrodes (Figure 5A). These

electrodes were grouped by region ». Also, are these ROI different from the previous ones? Please clarify for the reader.

6) Even though they are of great interest, I find the results and interpretations of the functional connectivity analysis rather ambitious in the space domain, given the fact that it is based on average magnitude-squared coherence. In scalp (referenced) EEG, such measurement is very prone to artefacts from volume conduction. I would recommend to reprocess data using the imaginary part of coherence for scalp level analysis (see Nolte et al., 2004), or applying spatial filtering (such as Laplacian) prior to coherence computation, or going into source space analysis (even in absence of individual MRI, i.e. on a brain template). Also, the authors should show the connectivity pattern at rest, i.e. during baseline (it is stated in the methods, but no results are discussed afterwards - or is it included somehow in the statistical model? Please clarify).

7) The authors consider that the temporary vestibulo-perceptual "neglect" induced by asymmetric vestibular stimulation could be mediated by alpha rhythms and frontoparietal attentional networks. This is consistent with previous data showing that vestibular processing as well as self-motion perception are related to modulation of alpha power over the left and right temporoparietal regions (Gale et al., 2016 [cited] but see and cite also Gutteling & Medendorp, 2016). This should also be discussed in relation to recent research showing an increase in alpha activity when self-motion perception is visually induced - vection - (Harquel et al., 2020; McAssey et al., 2020). This is of particular interest as the level of alpha activity in the vestibular cortical area varies with visually induced self-motion. These later results deserve to be put into perspective with the vestibule-perceptual "neglect" vs EEG modulation reported here.

Minors

8) Please specify what was the a priori rule for defining nystagmus in EOG data.

9) The description of the EEG preprocessing and analysis is lacking some details. Please specify the average +/- std length of the resulting epochs (is there a difference between cycles? This is important in respect of spectral analysis like FFT). Please specify ROI by depicting electrodes defining each ROI (on a blank topoplot figure for example).

10) The Figure 1Cii referenced in page 7 is missing.

11) For clarity, the authors should not overlay electrodes positions using black dots in topographic views, since it is hiding the electrical field (and given the fact that the EEG cap is in the 10-20 standard system).

END OF COMMENTS

Confidential Review

22-Nov-2021

Dear Editor:

Thank you for an overall positive assessment of our manuscript and the useful comments by the reviewers. Below are our replies and changes implemented in the manuscript, in **blue font** and **red font**, respectively. We feel that this round of revision and responses has resulted in an improved manuscript.

EDITOR COMMENTS

Reviewing Editor:

Methods Details:

Please specify in methods when the subjects are required to report their orientations in the room, the rule for defining nystagmus in EOG data and EEG preprocessing and analysis, as well as relevant ethical information, according to the reviewers's suggestions.

REPLY: Thank you. Please see the relevant comments for implemented changes.

Comments to the Author:

The manuscript reports an interesting EEG correlates of vestibular asymmetric perception in healthy humans. Please extensively revise the manuscript according to the reviewers' suggestions.

Senior Editor:

Comments for Authors to ensure the paper complies with the Statistics Policy: Please extend the provision of raw data to the EEG-derived measures. At present, they are restricted to the behavioural data. It is appreciated that raw data have been provided in the supplemental materials. An indication of individual responses on the figures would, however, also be beneficial.

REPLY: Thank you for your comment. We attempted to add raw data points on graphs of EEG derived measures but found that doing so overwhelmed the figure to the point of making them illegible.

Comments to the Author:

It is noted that, "data were analysed using linear mixed-effects models (LMMs)". Please provide a more complete description of these models (rather than a generic reference to

the type of model), that includes specification of the manner in which the data from individual participants were included in the analysis design. In reporting the outcomes of these analyses, the degrees of freedom associated with each F ratio should be reported in every instance. It is also desirable that an appropriate corresponding measure of effect size (preferably including confidence intervals) be provided in every instance.

REPLY: Thank you for requesting clarification on the specification of the LMMs used. We utilised a random slope-intercept model that allowed for heterogenous residual variances. This model included the fixed effects of interest – which varied depending on which outcome measure we were analysing, but could include condition, region, or frequency, as a few examples – as well as random effects. The random slope-intercept effects modelled the random deviations of participants from the fixed intercept; this model allowed these random deviations to be modelled differently for each condition and allowed residual variance to differ between various levels of a fixed factor, such as condition. More detailed explanations of these models can be found in the textbook *Linear Mixed Models: A Practical Guide Using Statistical Software*, 2nd Ed. By West, Welch, and Galecki (2015). Increased specification of this model has been added to the first paragraph of the *Statistical Analysis* section of the Methods, as specified below.

ADDITION (pg 9, line 3):

“Unless otherwise specified, data were analysed using linear mixed-effects models (LMMs), which modelled both fixed-effect parameters associated with covariates of interest (e.g., sex, treatment, experimental condition, etc.) and random effects, which model random deviance of outcome measures from fixed factors of interest, thus improving the predictive power of a model (West *et al.*, 2015). Unlike conventional linear models, LMMS are also able to include subjects with missing data, thereby allowing us to maximise use of data even after outlier removal and epoch rejection. We used LMMs that included both fixed effects (e.g., region, condition, etc.) and random effects associated with the individual deviations from the fixed intercept for each condition, as well as allowing for heterogenous residual variances (West *et al.*, 2015).”

Degrees of freedom have been reported where applicable in the text.

It appears that VOR gain was calculated for only 14 participants, whereas the Time-Based Spectral Analysis was based on 25 participants. Please provide a justification for the use of distinct pools of participants. Please provide evidence to support the assumption that the pattern of outcomes obtained for the 14 participants in respect of the VOR response could be generalised to the larger pool of 25 participants. In this context, consider that the "not significant" effect of frequency would almost certainly have been "statistically significant" (i.e., with a conventional NHST alpha level of 0.05), if the sample size had been 25 rather than 14. As alluded to above, the provision of

effect size estimates will provide the reader with a basis upon which to judge whether an effect was likely to have been present. Relatedly, what were the patterns of outcomes for the subset of 14 participants, in respect of the EEG- derived measures?

REPLY: We apologize for the lack of clarity around the VOR analysis. The smaller sample size for VOR analysis was due to poor EOG recordings for several participants, due to the presence of EMG, blinks, and roving eye movement artefacts. It is for this reason that, almost without exception, EOG recordings of vestibular responses with eyes closed have been abandoned. For more than 30 years, recordings have been conducted in total darkness with eyes open. In our case, it was necessary to record EOG with eyes closed because the room could not be completely darkened as the experimenter needed minimal light to move the chair.

However, we did not find this to be an issue of great concern. The VOR and behavioural response to this paradigm has been well described in previous studies (Pettorossi *et al.* 2013; Panichi *et al.* 2017a, 2017b; Faralli *et al.* 2021). We collected these measures to simply validate that behavioural adaptation had occurred and confirm that asymmetrical rotation induced distinct adaptive effects for reflexive and perceptual responses. As such, the VOR findings are ancillary to the main findings of interest, i.e., the EEG markers, and are not discussed in direct relation to other outcome measures. Statements to this effect have been added to the Methods (*EEG Data Acquisition*), Results (first paragraph), and Discussion (first paragraph) sections.

ADDITION (pg 7, line 13):

“As an ancillary measure to verify that participants were adapting as previously described (Pettorossi *et al.*, 2013; Panichi *et al.*, 2017a, 2017b; Faralli *et al.*, 2021), eye movements were recorded simultaneously (bi-temporal EOG) with electrodes placed on the lateral canthi of each eye. Note that while EOG is not conventionally recorded with eyes closed, as it increases EMG and roving eye movement artefacts, it was necessary participants close their eyes, recalling that the room was not completely darkened so that the experimenter could move the chair.”

ADDITION (pg 10, line 13):

“Additionally, as quality of EOG recordings was extremely variable both within and between participants, only clearly defined nystagmus was measured to calculate VOR gain, resulting in eleven participants with poor quality eye traces (likely due to roving eye artefacts) being excluded from analysis of the VOR response ($n = 14$ subjects). Although only a small pool was available for analysis, recall that the purpose of measuring VOR gain was simply to confirm adaptation had occurred; this response has already been thoroughly characterised in previous work (Pettorossi *et al.*, 2013; Panichi

et al., 2017a, 2017b; Faralli et al., 2021) and as such was not a main focus of this study.”

ADDITION (pg 15, line 2):

“Our results replicated previous findings that asymmetric stimulation can induce acute perceptual vestibular neglect and that this is not reflected in brainstem reflex mechanisms such as VOR (Pettorossi *et al.*, 2013; Panichi *et al.*, 2017a, 2017b; Faralli *et al.*, 2021). As these psychophysical and reflexive responses have already been thoroughly described in the aforementioned studies – and given that the main aim of the present study was to determine the EEG markers of the perceptual adaptation – we will briefly comment on the behavioural importance of dissociating the two responses but will not discuss our findings around these measures further.”

REFEREE COMMENTS

Referee #1:

In the current study, Cooke *et al.*, studied EEG correlate of vestibular asymmetric perception illusion when conflict fast and slow sinusoidal oscillations were provided. In the behaviour, subjects typically ignored the slow oscillation by reporting the fast motion phase, whereas VOR typically faithfully reflects both motion phases. The authors did not find correlates of EEG with the perceptual error, unfortunately, however, there is a trend that during conflict sinusoidal asymmetric oscillations, there is a desynchronisation in the frontal cortex, and a resynchronization in the occipital cortex, indicating the frontal-parietal network is inhibiting some conflict signals during the asymmetric oscillation conditions. The motivation of the study is nice. The study is carefully conducted. The results are clearly presented. I only have a few comments.

1. The method is not clear enough about when the subjects are required to report their orientations in the room. Is it always within one cycle, or it could be more than one cycle? The landmark is 45 degrees apart. It seems a bit coarse. One cannot be more fine steps?

REPLY: We acknowledge that the resolution used for incremental steps between landmarks was indeed large. However, our results show, after 16 cycles of asymmetric rotation, average error was much larger than 45° (additionally, only one of the participants who adapted reported total position error at this minimal threshold; other

errors reported were substantially greater) so the 45° step assures us that adaptation occurred. Below this value, our experience (Pettorossi *et al.* 2013; Panichi *et al.* 2017a, 2017b; Faralli *et al.* 2021) tells us that there was no adaptive effect after several cycles of oscillation. A statement summarising this has been included in the *Experimental Protocol* section of the Methods.

ADDITION (pg 6, line 20):

“Although this resolution was somewhat coarse, previous work (Pettorossi *et al.*, 2013) indicated that 3-4 oscillations usually elicits an error in the range of 30-40°, with additional cycles demonstrating a cumulative error. With participant’s experiencing 16 cycles, it followed that if they didn’t report an error of at least 45°, adaptation had not occurred.”

More importantly, as stated in a response above, the main purpose of collecting and reporting position error was to simply confirm, from a behavioural point of view, that adaptation has occurred. The psychophysics of this adaptive response have already been described in detail in previous work (Pettorossi *et al.* 2013; Panichi *et al.* 2017a, 2017b; Faralli *et al.* 2021); the novelty here lies in the associated EEG markers. More precision may have allowed for us to define a quantitative relationship between EEG activity and the extent of adaptation, but there was concern that additional measuring devices would have introduced electrical interference with the EEG signal, as well as produce motor-related signals that would have been difficult to dissociate from the markers of vestibular adaptation we were interested in. This may be addressed in future studies.

Regarding your comment about when subjects were required to report their perceived orientations, we apologise for the lack of the clarity. Subjects were asked to report their final perceived orientation once per condition at the end of the 16 cycles (and after the two-minute post-adaptation resting state recording). This has been clarified in the Methods section (*Experimental Protocol*).

ADDITION (pg 7, line 1):

“Finally, at the culmination of the recording session, participants were asked to state their final perceived facing.”

2. In the symmetric condition, the errors could be very large for the slow motion condition. For example, as shown in Figure 2A, some subjects show error more than one cycle (>360 deg). In these cases, did the subjects really understand the task, or the errors really reflect their sensory error?

We appreciate your comment regarding subject performance. The larger perceptual error noted in the lower frequency symmetric stimulus was indeed unexpected. However, this may be attributed to that fact that the overall frequency of the stimulus was toward the lower edge of vestibular sensitivity; direction detection thresholds for yaw rotation increase at frequencies of ≤ 0.2 Hz (Grabherr *et al.*, 2008; Carriot *et al.*, 2014; Mallery *et al.*, 2014). As such, it is possible that some participants struggled to discriminate between leftward and rightward motion, and thus perceived themselves to be rotating in one direction. Note that this explanation is distinct from that of the behavioural adaptation that occurs during asymmetric stimulation, which we propose is an effect of attentional reorienting to the more relevant stimulus (i.e., the FHC). This distinction is supported by the differing trends of alpha rhythm resynchronisation during symmetric versus asymmetric stimulation.

ADDITION (pg 15, line 10):

“Firstly, it was noted that there was a considerable degree of error in the low-frequency symmetric condition, although this error was noted in both directions. We attribute this to the stimulus frequency; direction detection thresholds rise below 0.2 Hz rotations (Grabherr *et al.*, 2008; Carriot *et al.*, 2014; Mallery *et al.*, 2014), and as such some participants may have perceived all motion in one direction. This is still behaviourally distinct, however, from the adaptation noted with the asymmetric stimulus, the proposed purpose of which is discussed in greater detail below.”

3. All the figures are not clear enough. They should have higher resolution and quality.

REPLY: Thank you for your comment. We have recreated figures at higher resolution.

Referee #2:

Thank you for submitting your manuscript to The Journal of Physiology.

You must begin the Methods section with the sub-heading "Ethical approval". The relevant information required in this section is already provided by the authors in the opening paragraph.

REPLY: A section titled *Ethical Approval* (pg 4, line 16) has been added to the beginning of the Methods section, in line with journal requirements.

Please refer to "participants" rather than "subjects".

REPLY: All instances of the term "subjects" have been replaced with the term "participants". First instance of this replacement is has been indicated in the revised text (pg 4, line 23); the remaining instances are not highlighted.

It would be helpful to include a statement that there were no adverse outcomes and no lasting disorientation in the participants at the close of the study.

REPLY: A statement to this affect has been added at the end of the *Experimental Protocol* section of methods.

ADDITION (pg 7, line 8): "No adverse outcomes nor persistent sensations of disorientation were reported by participants after study completion."

Referee #3:

In the present manuscript, the authors first replicated previous findings that asymmetric body stimulation (rotation along the z-axis) induces perceptual errors (so-called "vestibular neglect") although it does not impact brainstem reflex mechanisms such as the VOR. Secondly, the authors showed that the alpha frequency band was related to the adaptive process induced by asymmetric rotation. Although the current experiment is of interest and appears to be well conducted, I have some concerns that should be taken into account.

Majors

1) Looking at Figure 1, the traces of the samples reflect the fact that the chair is manually operated, as the profiles of these traces do not correspond to pure sinusoids, which is particularly visible with regard to the velocity. The fact that the chair is manually operated is understandable if one considers that the authors wish to limit the electrical noise in a Faraday cage. However, if one considers that the vestibular system (here the semi-circular canals) is sensitive to accelerations, one can assume that the acceleration profiles are even further away from pure sinusoids. Considering that the traces reported in the figure are probably among the most presentable traces, can the authors reassure the reader about the quality of the stimulus considering that the experimental aim is to evaluate a perceptual dimension (vestibular neglect) that arises from an acceleration

sensitive system.

REPLY: Thank you for your comment. Stimuli were hand delivered always by the same experimenter (the first author) who is an ex-professional dancer with ample experience in rhythmic, repetitive movements. Further, recall the manual oscillation was guided by a metronome and an amplitude calibrated visual display. We instrumented a training period until we were satisfied that the rotations were as intended before starting the experiments. Once the experiments commenced, traces were always visually inspected at the end of each session to ensure consistency and regularity of the stimulus profile. In addition, at the end of the project we randomly selected 10 participants and overlaid their stimuli to visualise discrepancies between traces. We also performed a Fast Fourier Transform (FFT) of the stimulus traces, overlaying the FFT results as well. These data are shown in **Supplementary Figures 1 and 2**, respectively, showing minimal variability in the shape and frequency content of the stimuli. A statement to the above effect was added to the *Experimental Design* section of the Methods, as stated below.

ADDITION (pg 5, line 28): “The experimenter practised rotating the chair at these frequencies at length prior to beginning the experiment; traces were examined at the end of each session to ensure regularity and consistency of the stimulus. In addition, at the conclusion of the study, the recorded stimulus traces of 10 randomly selected participants were overlaid to check for discrepancies (**Supplementary Figure 1**). A Fast Fourier Transform was also performed on the traces, and the results overlaid (**Supplementary Figure 2**). Both analyses showed minimal variability in the shape and frequency content of the stimuli, thus demonstrating the stimulation was consistent across participants.”

2) The authors report a dissociation between the VOR, which is unaffected by stimulus asymmetry, which is also a direct measure of vestibular function, and perception, which is particularly sensitive to this asymmetry and is thought to reflect a high level of vestibular interpretation. It turns out that the rotation of the participants to the right and to the left stimulates of course the vestibular system, but does also stimulate the proprioceptive system (through pressures on the body surface that differ according to the speed and acceleration of the body in space). In this respect, one may wonder whether the dissociation between VOR and perception (consider as vestibular neglect) in response to stimulus asymmetry does not only vestibular signals involvement in reflex activities (VOR) but vestibular as well as proprioceptive signals for perception. This is should definitely be discussed.

REPLY: We thank you for highlighting a potential confound in isolating vestibular input. A small proprioceptive component during asymmetric rotation signal may be elicited. However, given the transfer characteristics of the vestibular and proprioceptive systems it would be unlikely that any putative proprioceptive input would have a significant influence on our asymmetric adaptation paradigm. In a study by Mergner et al (2001) and in my own (Nakamura and Bronstein, 1995; e.g., see Figure 6 there) the gain of the vestibulo-perceptual response increases as a function of stimulus frequency/velocity, in contrast to the gain of the proprioceptive perceptual response which remains essentially unchanged. This means that any increased proprioceptive input brought about by the higher acceleration present during the fast hemicycle would be relatively less efficiently sensed by the proprioceptive than by the vestibular system. Additionally, asymmetrical body rotations with passive head rotation were previously examined to look for the role of neck proprioception on movement perception and found to not impact the perceptual adaptation (Panichi *et al.*, 2011). In addition, recent published articles examined this paradigm in vestibular neuritis (Panichi *et al.*, 2017a) and Meniere's disease (Faralli *et al.*, 2021) patient populations, showing that vestibular impairment significantly alters this adaptive perceptual response. Thus, the vestibular system is the system most involved in the adaptive phenomenon. This has been added to the *Potential Role of Subcortical Circuits* section of the Discussion.

ADDITION (pg 19, line 27):

“A separate consideration is the possible influence of low-level proprioceptive signals on the adaptive response. Although every measure was taken to restrict proprioceptive input during the experimental set-up, it is possible there was a proprioceptive contribution to the perceptual response. However, it is unlikely that this had a significant effect on the adaptation noted. Increased stimulus frequency has been shown to increase the gain of the vestibulo-perceptual response, but not the proprioceptive response (Nakamura & Bronstein, 1995; Mergner *et al.*, 2001), meaning the increased acceleration of the FHC would be sensed less efficiently by the proprioceptive system relative to the vestibular system. Further evidence of this is provided by previous work which found that passive head deviations during asymmetric rotations had no impact on movement perception (Panichi *et al.*, 2011). Additionally, studies of this paradigm in vestibular neuritis (Panichi *et al.*, 2017a) and Meniere's disease (Faralli *et al.*, 2021) populations have shown that vestibular impairment alters the adaptive response noted in healthy individuals. Taken together, these findings, along with those in the present study, support that the adaptation is due to altered vestibular perception.”

3) The definition of alpha rhythm 8-14 Hz is atypical and does not fit to the recommended standard (please see table 1 of Babiloni et al. 2020 for a recent review),

which is important for comparing studies. The upper bound is in most cases equal to 12 Hz and should never be higher than 13 Hz. The authors should define the alpha bandwidth in respect of these international guidelines, or at least discuss the reasons of this discrepancy.

REPLY: We acknowledge that the upper range of alpha frequency is often defined at 12 or 13 Hz, particularly when averaging across an entire band. Alpha power used for our statistical analyses, however, was taken from the individual alpha frequency (IAF) for each participant. This measure has historically been determined as the peak between 6 and 14 Hz (Klimesch, 1999) and studies that utilise IAF-defined bands have extended the upper limit to 14 Hz (Babiloni *et al.*, 2020b). This has been clarified in the *EEG Analysis* section of the Methods.

ADDITION (pg 8, line 6):

“For the alpha frequency band, power was measured at individual alpha frequency (IAF). The upper bound of alpha frequencies is usually defined at 12 or 13 Hz; however, IAF has been found to go up to 14Hz (Klimesch, 1999; Babiloni *et al.*, 2020b, 2020a), hence the larger band.”

That said, the average IAF across all participants (averaged across all electrodes) was 10.47 Hz ($SD = 0.85$ Hz), with the minimum IAF measured at 8.43 Hz and the maximum at 12.68 Hz. These descriptive statistics have been added to the *Time-Based Spectral Analysis* section of the results.

ADDITION (pg 12, line 31):

“Individual alpha frequency (IAF) – the point at which power was measured – averaged (SD) at 10.47 Hz (0.85 Hz) across all regions, with values ranging from 8.43 to 12.68 Hz.”

4) The Time-Based Spectral Analysis section begins with several visual observations and (what seems to be) arbitrary choices. First, the authors stated that "Average normalised power of each cycle was also examined and confirmed the previously noted observations, namely that there was a greater suppression of alpha power in the symmetric condition relative to the asymmetric condition and that the parietal, occipital, prefrontal, and frontal regions showed this response more clearly (Figure 3B). There was no observable lateralisation of this response." The authors should run statistical analyses in order to support these observations (i.e., sign. difference from baseline, lateralization), prior to the ROI creation ("As a result, power was averaged across the electrodes in these four regions, henceforth referred to as regions of interest (ROIs), for further analysis »). Second, the authors should justify the choice for plotting O2 for Fig 3A. Also, why not plot the last cycle in Fig 3A?

REPLY: Thank you for asking for additional clarification. We viewed the visual observations as an integral part of the data analysis process. We trust that the Referee agrees that visual examination of raw physiological data has always been utilised to get a sense of possible trends in the data and inform the direction of statistical analysis.

ADDITION (pg 12, line 6):

“Close visual examination of spectrograms at various electrodes was utilised to roughly identify trends in the data. This preliminary analysis yielded several important observations.”

Part of the reason for choosing to use the spectrograms of the O2 electrode in the manuscript is that it demonstrates one of our initial visual observations quite clearly, namely that alpha power (which is very visible as the bright yellow band in the spectrograms, particularly in occipital electrodes) was greater at baseline than during the adaptation period, for both the symmetric and asymmetric conditions.

ADDITION (pg 12, line 11):

“This can be clearly observed in the occipital region, which is known to receive substantial propagation of alpha rhythms from subcortical generators (Halgren *et al.*, 2019); the O2 electrode, depicted in **Figure 3A**, demonstrates this well.”

Regarding the comparison between baseline measures and those during the *Adaptation* period, the analysis of the time-based spectral data uses normalised power values, i.e., raw power (averaged across an epoch; the epoch corresponding to one full oscillation of the chair) during the *Adaptation* period divided by raw power (averaged across a corresponding epoch) during the *Baseline* period. This normalisation served two purposes; first it reduces the high degree of variance in raw power values both within and between subjects. Second, it directly quantifies the relationship between power at baseline versus power during adaptation, i.e., a value less than 1 would indicate power during the *Adaptation* period is less than power at baseline (also termed here as desynchronisation), while a value greater than 1 indicates power is greater than at baseline (increased synchronisation). The relationship between power and synchronicity is described in greater detail by Pfurtscheller and colleagues (Pfurtscheller & Lopes Da Silva, 1999; Pfurtscheller, 2001). We used a simplified version of the calculations specified there. As such, the signal difference from baseline is already included in the statistical model for all spectral analysis (time-based and steady state).

ADDITION (pg 8, line 10):

“To reduce inter-individual variance of EEG, the *Adaptation* and *Post-Adaptation* recordings were normalised (divided) by the baseline recording. This normalisation

process additionally provided an indication of the relationship between raw power during the *Adaptation* period versus the *Baseline* period. Specifically, it indicates if power is decreased during the *Adaptation* period relative to baseline, (note: as this reduction results from decreased synchrony of the underlying neural population, it is termed desynchronisation) or increased (increased synchrony; termed synchronisation) (Pfurtscheller & Lopes Da Silva, 1999; Pfurtscheller, 2001). Thus, normalised power values greater than 1 indicated synchronisation, values less than 1 indicated desynchronisation.”

Selection of the ROIs was largely determined *a priori* from initial literature review, which, as summarised in the introduction, highlighted the role of the parieto-occipital cortices in vestibular processing (Lopez & Blanke, 2011; Gale *et al.*, 2016) as well as the role of frontoparietal attentional networks in error processing and sensory adaptation (Corbetta & Shulman, 2002; Petersen & Posner, 2012; Scolari *et al.*, 2015). Selection of ROIs was additionally determined from our initial visual observation of the averaged spectrogram data (i.e., the headplots shown in **Figure 3B**). Rationale for ROI selection and the list of electrodes included in each ROI have been specified in the *Time-Based Spectral Analysis* section of the Results.

ADDITION (pg 12, line 19):

“Regions of interest were partially selected *a priori*, based on previous work postulating the role of the parieto-occipital cortices in vestibular processing (Lopez & Blanke, 2011; Gale *et al.*, 2016) as well as the role of frontoparietal attentional networks in error processing and sensory adaptation (Corbetta & Shulman, 2002; Petersen & Posner, 2012; Scolari *et al.*, 2015). The decision to focus on these regions was further confirmed through visual examination of the average normalised power of each spectrogram (**Figure 3B**), which aligned the previously noted observations; there was a greater suppression of alpha power in the symmetric condition relative to the asymmetric condition and this was largely reflected in the parietal (P7, P3, Pz, P4, and P8 electrodes), occipital (O1, Oz, and O2 electrodes), prefrontal (Fp1, Fpz, and Fp2 electrodes), and frontal (F7, F3, Fz, F4, and F8 electrodes) regions (**Figure 3B**).”

The last cycle has been plotted in Figure 3A.

5) The same goes for the ROI definition of Steady state spectral power analysis: "Head plots of steady state power showed no lateralisation either the Adaptation or Post-Adaptation in any frequency band, but several regions appeared to differ between conditions, including frontal, prefrontal, parietal, centroparietal, and occipital electrodes (Figure 5A). These electrodes were grouped by region ». Also, are these ROI different from the previous ones? Please clarify for the reader.

REPLY: We apologise for the lack of clarity around the definitions of ROIs. The ROIs used during the steady state analysis are identical to those described earlier and were chosen in an identical manner to that described in the previous comment. The only distinction is that centroparietal electrodes, which were observed to differ between conditions, were added as a ROI. This has been clarified in the *Steady State Spectral Power* section of the Results.

ADDITION (pg 13, line 26):

“As in the previous section, visual examination was conducted on normalised steady state power head plots to determine ROIs to analyse. No lateralisation was observed in either the *Adaptation* or *Post-Adaptation* in any frequency band, but several regions (most of which were predicted to be relevant *a priori*) appeared to differ between conditions, including frontal, prefrontal, parietal, centroparietal, and occipital electrodes (**Figure 5A**). These electrodes were grouped by region (same ROIs as before, but with the centroparietal electrodes – CP5, CP1, CP2, AND CP6 – being included)”

6) Even though they are of great interest, I find the results and interpretations of the functional connectivity analysis rather ambitious in the space domain, given the fact that it is based on average magnitude-squared coherence. In scalp (referenced) EEG, such measurement is very prone to artefacts from volume conduction. I would recommend reprocessing data using the imaginary part of coherence for scalp level analysis (see Nolte et al., 2004), or applying spatial filtering (such as Laplacian) prior to coherence computation or going into source space analysis (even in absence of individual MRI, i.e., on a brain template). Also, the authors should show the connectivity pattern at rest, i.e., during baseline (it is stated in the methods, but no results are discussed afterwards - or is it included somehow in the statistical model? Please clarify).

REPLY: We acknowledge that volume conduction is typically an issue when conducting coherence analysis. This is more problematic, however, when conducting experiments with a between-subjects design (i.e., different subject groups). As we were comparing the influence of different stimuli on the same group of participants (i.e., a within-subjects design), factors that usually influence volume conduction, such as tissue thickness, are inherently controlled by the study design.

Another factor usually related to volume conduction is proximity of electrodes; however, the edges (connections) identified in our network analysis are mostly comprised of distant nodes (electrodes), namely (pre)frontal and parieto-occipital electrodes. Furthermore, the electrodes between these regions do not show significant connectivity (except for CP1, CPz, and Pz in the beta frequency band; this will be discussed momentarily) and connections are fairly lateralised; if volume conduction was driving the

results, we would expect to see multiple connections between neighbouring electrodes on both sides of the scalp. Given this, we argue that our results are valid and are largely indicative of stimulus-related changes in network connectivity.

That said, we did reprocess data using imaginary coherence (IC) to examine the cluster of neighbouring electrodes mentioned above, which may have been more influenced by volume conduction. The reanalysis found a significant difference in network strength between conditions during the adaptation period for beta frequency ($p = 0.039$, see Figure 6B). Although the connections identified differed from those identified in the previous analysis of magnitude-squared coherence (MSC), more than 50% of the electrodes in the MSC-derived network were also involved in the IC-derived network. Most interestingly, inter-electrode connectivity findings were reproduced for the same cluster of midline “z” electrodes. While unexpected to find this pattern maintained, there is strong evidence that midline electrodes pick up late vestibular activity (Todd *et al.*, 2014; Nakul *et al.*, 2021). The above has been added to the results and discussion where applicable.

ADDITION (pg 14, line 22):

“Concern around the results being attributable to volume conduction (VC) was minimal, given that we utilised a within-subjects design (i.e., many factors influencing VC were inherently controlled for), and given the distance between nodes (electrodes) comprising edges (connections). However, there was a cluster of neighbouring electrodes identified in the beta network (CP1, CPz, and Pz) that were identified as potentially susceptible to VC. To further investigate, data were reprocessed using imaginary coherence (Nolte *et al.*, 2004). This analysis identified a network of 12 connections in the beta band only ($FWER-p = 0.039$), primarily between bilateral frontocentral and centroparietal electrodes (**Figure 6B**).”

ADDITION (pg 17, line 30):

“The results of the imaginary coherence analysis were unexpected, as the network identified was comprised almost exclusively of fronto-central and centroparietal connections. However, it has been theorised that midline regions can reflect late latency vestibular signals (Todd *et al.*, 2014; Nakul *et al.*, 2021), which may reflect some of the bottom-up feed-forward mechanisms discussed previously.”

Finally, regarding networks at baseline, thank you for identifying the lack of clarity in the methods. The comparison made in these analyses is between the asymmetric and symmetric condition (which serves as a control); this comparison was run *separately* for the baseline, adaptation, and post-adaptation periods, as these were all distinct states

(i.e., we would not expect to find the same network at baseline as we would at during specific task). This has been clarified in the *statistical Analysis* section of the Methods.

ADDITION (pg 10, line 21):

“Average magnitude-squared coherence was used as the measure of connectivity for our networks, which directly compared the asymmetric and asymmetry, separately for each period (*Baseline, Adaptation, and Post-Adaptation*).”

7) The authors consider that the temporary vestibulo-perceptual "neglect" induced by asymmetric vestibular stimulation could be mediated by alpha rhythms and frontoparietal attentional networks. This is consistent with previous data showing that vestibular processing as well as self-motion perception are related to modulation of alpha power over the left and right temporoparietal regions (Gale et al., 2016 [cited] but see and cite also Gutteling & Medendorp, 2016). This should also be discussed in relation to recent research showing an increase in alpha activity when self-motion perception is visually induced -vection- (Harquel et al., 2020; McAssey et al., 2020). This is of particular interest as the level of alpha activity in the vestibular cortical area varies with visually induced self-motion. These later results deserve to be put into perspective with the vestibule-perceptual "neglect" vs EEG modulation reported here.

REPLY: Thank your for suggesting these papers. They indeed align with our results nicely. They have been incorporated in the Discussion as follows

ADDITION (pg 15, line 31):

“...specifically that desynchronisation and subsequent rebound of alpha power differs in response to asymmetric versus symmetric stimulation, particularly in the prefrontal and occipital regions, where the resynchronisation is accelerated and diminished, respectively. Body motion has been shown to correspond with a reduction of alpha power in central parietal, and lateral parieto-temporal areas (Gutteling & Medendorp, 2016). Gale *et al.* (2016) additionally showed that a continuously perceived vestibular stimulation is associated with suppression of alpha over the parietal cortex. Our findings show the same initial reduction, which then decreases over the course of continual stimulation. This trend has been noted in the literature previously; a relative increase after initial reduction was noted by Gale et al. (2016), as well as in separate studies using section to induce motion perception (Harquel *et al.*, 2020; McAssey *et al.*, 2020).

Our findings, along with previous work, suggest that the initial alpha reduction indicates neural pathways are primed to detect and process incoming information, but then are fine-tuned to process salient information. This supposition, which is also discussed in similar work (Pettorossi *et al.*, 2013; Harquel *et al.*, 2020) is supported by work on the role of enhanced alpha as an inhibitory gating mechanism...”

Minors

8) Please specify what was the a priori rule for defining nystagmus in EOG data.

REPLY: Vestibular nystagmus was identified when a consistent slow phase eye movement of at least 4 deg/s was followed by an oppositely directed fast phase. A sentence clarifying this criterion has been added to the *Statistical Analysis* section of the Methods.

ADDITION (pg 9, line 17):

“Nystagmus was identified by trained experimenters (first, third, and sixth authors) as characteristic jerk nystagmus (Serra & Leigh, 2002), i.e., slow drift of at least 4°/s from a fixation point immediately followed by rapid corrective motion in the opposite direction. Traces that did not demonstrate clear nystagmus were rejected.”

9) The description of the EEG preprocessing and analysis is lacking some details. Please specify the average +/- std length of the resulting epochs (is there a difference between cycles? This is important in respect of spectral analysis like FFT). Please specify ROI by depicting electrodes defining each ROI (on a blank topoplot figure for example).

REPLY: We apologise for the lack of clarification regarding epoch length. Epochs were all identical in length, measured as 5 seconds (asymmetric) or 8 seconds (symmetric) from peak displacement of the chair. A sentence clarifying this has been added to the Methods.

ADDITION (pg 8, line 21):

“Epochs were 5s (asymmetric) or 8s (symmetric) in length, measured from peak displacement of each cycle of the chair position traces.”

A blank topoplot with regions identified was added to Figure 4.

10) The Figure 1Cii referenced in page 7 is missing.

REPLY: Thank you for identifying this error. The erroneous reference to a non-existent figure has been removed.

11) For clarity, the authors should not overlay electrodes positions using black dots in topographic views, since it is hiding the electrical field (and given the fact that the EEG cap is in the 10-20 standard system).

REPLY: Thank you for this suggestion. Headplots have been reformatted without the electrode positions and replaced in the applicable figures.

Dear Professor Bronstein,

Re: JP-RP-2022-282470R1 "Electroencephalographic Response to Transient Adaptation of Vestibular Perception" by Josephine I Cooke, Onur Guven, Patricia Castro Abarca, Richard T Ibitoye, Vito E Pettorossi, and Adolfo M. Bronstein

Thank you for submitting your manuscript to The Journal of Physiology. It has been assessed by a Reviewing Editor and by 2 expert Referees and I am pleased to tell you that it is almost ready for acceptance. Before formal acceptance, however, please attend to the minor comments below (one typo, and please provide figure legends).

The reports are copied at the end of this email. Please address all of the points and incorporate all requested revisions.

NEW POLICY: In order to improve the transparency of its peer review process The Journal of Physiology publishes online as supporting information the peer review history of all articles accepted for publication. Readers will have access to decision letters, including all Editors' comments and referee reports, for each version of the manuscript and any author responses to peer review comments. Referees can decide whether or not they wish to be named on the peer review history document.

Authors are asked to use The Journal's premium BioRender (<https://biorender.com/>) account to create/redraw their Abstract Figures. Information on how to access The Journal's premium BioRender account is here: <https://physoc.onlinelibrary.wiley.com/journal/14697793/biorender-access> and authors are expected to use this service. This will enable Authors to download high-resolution versions of their figures. The link provided should only be used for the purposes of this submission. Authors will be charged for figures created on this premium BioRender account if they are not related to this manuscript submission.

I hope you will find the comments helpful and have no difficulty returning your revisions within 7 days.

Your revised manuscript should be submitted online using the links in Author Tasks Link Not Available.

Any image files uploaded with the previous version are retained on the system. Please ensure you replace or remove all files that have been revised.

REVISION CHECKLIST:

- Article file, including any tables and figure legends, must be in an editable format (eg Word)
- Abstract figure file (see above)
- Statistical Summary Document
- Upload each figure as a separate high quality file
- Upload a full Response to Referees, including a response to any Senior and Reviewing Editor Comments;
- Upload a copy of the manuscript with the changes highlighted.

- A potential 'Cover Art' file for consideration as the Issue's cover image;
- Appropriate Supporting Information (Video, audio or data set https://jp.msubmit.net/cgi-bin/main.plex?form_type=display_requirements#supp).

To create your 'Response to Referees' copy all the reports, including any comments from the Senior and Reviewing Editors, into a Word, or similar, file and respond to each point in colour or CAPITALS and upload this when you submit your revision.

I look forward to receiving your revised submission.

If you have any queries please reply to this email and staff will be happy to assist.

Yours sincerely,

EDITOR COMMENTS

Reviewing Editor:

Please correct the typo pointed by Reviewer #1 and provide a caption to each figure.

The figure legends should be included in the article (Word) file.

REFEREE COMMENTS

Referee #1:

The authors have addressed my comments in thoroughly, and I do not have any further comments.

Referee #3:

The authors answered satisfactorily all my concerns.

There is only one typo that should be corrected: Page 16 line 8 "vection" instead of "section".

END OF COMMENTS

1st Confidential Review

06-May-2022

Dear Editor:

Thank you for an overall positive assessment of our manuscript and the useful comments by the reviewers. Below are our replies and changes implemented in the manuscript, in **blue font** and **red font**, respectively. We feel that this round of revision and responses has resulted in an improved manuscript.

EDITOR COMMENTS

Reviewing Editor:

Methods Details:

Please specify in methods when the subjects are required to report their orientations in the room, the rule for defining nystagmus in EOG data and EEG preprocessing and analysis, as well as relevant ethical information, according to the reviewers's suggestions.

REPLY: Thank you. Please see the relevant comments for implemented changes.

Comments to the Author:

The manuscript reports an interesting EEG correlates of vestibular asymmetric perception in healthy humans. Please extensively revise the manuscript according to the reviewers' suggestions.

Senior Editor:

Comments for Authors to ensure the paper complies with the Statistics Policy: Please extend the provision of raw data to the EEG-derived measures. At present, they are restricted to the behavioural data. It is appreciated that raw data have been provided in the supplemental materials. An indication of individual responses on the figures would, however, also be beneficial.

REPLY: Thank you for your comment. We attempted to add raw data points on graphs of EEG derived measures but found that doing so overwhelmed the figure to the point of making them illegible.

Comments to the Author:

It is noted that, "data were analysed using linear mixed-effects models (LMMs)". Please provide a more complete description of these models (rather than a generic reference to

the type of model), that includes specification of the manner in which the data from individual participants were included in the analysis design. In reporting the outcomes of these analyses, the degrees of freedom associated with each F ratio should be reported in every instance. It is also desirable that an appropriate corresponding measure of effect size (preferably including confidence intervals) be provided in every instance.

REPLY: Thank you for requesting clarification on the specification of the LMMs used. We utilised a random slope-intercept model that allowed for heterogenous residual variances. This model included the fixed effects of interest – which varied depending on which outcome measure we were analysing, but could include condition, region, or frequency, as a few examples – as well as random effects. The random slope-intercept effects modelled the random deviations of participants from the fixed intercept; this model allowed these random deviations to be modelled differently for each condition and allowed residual variance to differ between various levels of a fixed factor, such as condition. More detailed explanations of these models can be found in the textbook *Linear Mixed Models: A Practical Guide Using Statistical Software*, 2nd Ed. By West, Welch, and Galecki (2015). Increased specification of this model has been added to the first paragraph of the *Statistical Analysis* section of the Methods, as specified below.

ADDITION (pg 9, line 3):

“Unless otherwise specified, data were analysed using linear mixed-effects models (LMMs), which modelled both fixed-effect parameters associated with covariates of interest (e.g., brain region, experimental condition, frequency, etc.) and random effects, which model random deviance of outcome measures from fixed factors of interest, thus improving the predictive power of a model (West *et al.*, 2015). Unlike conventional linear models, LMMS are also able to include subjects with missing data, thereby allowing us to maximise use of data even after outlier removal and epoch rejection. We used LMMs that included both fixed effects (e.g., region, condition, etc.) and random effects associated with the individual deviations from the fixed intercept for each condition, as well as allowing for heterogenous residual variances (West *et al.*, 2015).”

Degrees of freedom have been reported where applicable in the text.

It appears that VOR gain was calculated for only 14 participants, whereas the Time-Based Spectral Analysis was based on 25 participants. Please provide a justification for the use of distinct pools of participants. Please provide evidence to support the assumption that the pattern of outcomes obtained for the 14 participants in respect of the VOR response could be generalised to the larger pool of 25 participants. In this context, consider that the "not significant" effect of frequency would almost certainly have been "statistically significant" (i.e., with a conventional NHST alpha level of 0.05), if the sample size had been 25 rather than 14. As alluded to above, the provision of

effect size estimates will provide the reader with a basis upon which to judge whether an effect was likely to have been present. Relatedly, what were the patterns of outcomes for the subset of 14 participants, in respect of the EEG- derived measures?

REPLY: We apologize for the lack of clarity around the VOR analysis. The smaller sample size for VOR analysis was due to poor EOG recordings for several participants, due to the presence of EMG, blinks, and roving eye movement artefacts. It is for this reason that, almost without exception, EOG recordings of vestibular responses with eyes closed have been abandoned. For more than 30 years, recordings have been conducted in total darkness with eyes open. In our case, it was necessary to record EOG with eyes closed because the room could not be completely darkened as the experimenter needed minimal light to move the chair.

However, we did not find this to be an issue of great concern. The VOR and behavioural response to this paradigm has been well described in previous studies (Pettorossi *et al.* 2013; Panichi *et al.* 2017a, 2017b; Faralli *et al.* 2021). We collected these measures to simply validate that behavioural adaptation had occurred and confirm that asymmetrical rotation induced distinct adaptive effects for reflexive and perceptual responses. As such, the VOR findings are ancillary to the main findings of interest, i.e., the EEG markers, and are not discussed in direct relation to other outcome measures. Statements to this effect have been added to the Methods (*EEG Data Acquisition*), Results (first paragraph), and Discussion (first paragraph) sections.

ADDITION (pg 7, line 13):

“As an ancillary measure to verify that participants were adapting as previously described (Pettorossi *et al.*, 2013; Panichi *et al.*, 2017a, 2017b; Faralli *et al.*, 2021), eye movements were recorded simultaneously (bi-temporal EOG) with electrodes placed on the lateral canthi of each eye. Note that while EOG is not conventionally recorded with eyes closed, as it increases EMG and roving eye movement artefacts, it was necessary participants close their eyes, recalling that the room was not completely darkened so that the experimenter could move the chair.”

ADDITION (pg 10, line 13):

“Additionally, as quality of EOG recordings was extremely variable both within and between participants, only clearly defined nystagmus was measured to calculate VOR gain, resulting in eleven participants with poor quality eye traces (likely due to roving eye artefacts) being excluded from analysis of the VOR response ($n = 14$ subjects). Although only a small pool was available for analysis, recall that the purpose of measuring VOR gain was simply to confirm adaptation had occurred; this response has already been thoroughly characterised in previous work (Pettorossi *et al.*, 2013; Panichi

et al., 2017a, 2017b; Faralli et al., 2021) and as such was not a main focus of this study.”

ADDITION (pg 15, line 2):

“Our results replicated previous findings that asymmetric stimulation can induce acute perceptual vestibular neglect and that this is not reflected in brainstem reflex mechanisms such as VOR (Pettorossi *et al.*, 2013; Panichi *et al.*, 2017a, 2017b; Faralli *et al.*, 2021). As these psychophysical and reflexive responses have already been thoroughly described in the aforementioned studies – and given that the main aim of the present study was to determine the EEG markers of the perceptual adaptation – we will briefly comment on the behavioural importance of dissociating the two responses but will not discuss our findings around these measures further.”

REFEREE COMMENTS

Referee #1:

In the current study, Cooke *et al.*, studied EEG correlate of vestibular asymmetric perception illusion when conflict fast and slow sinusoidal oscillations were provided. In the behaviour, subjects typically ignored the slow oscillation by reporting the fast motion phase, whereas VOR typically faithfully reflects both motion phases. The authors did not find correlates of EEG with the perceptual error, unfortunately, however, there is a trend that during conflict sinusoidal asymmetric oscillations, there is a desynchronisation in the frontal cortex, and a resynchronization in the occipital cortex, indicating the frontal-parietal network is inhibiting some conflict signals during the asymmetric oscillation conditions. The motivation of the study is nice. The study is carefully conducted. The results are clearly presented. I only have a few comments.

1. The method is not clear enough about when the subjects are required to report their orientations in the room. Is it always within one cycle, or it could be more than one cycle? The landmark is 45 degrees apart. It seems a bit coarse. One cannot be more fine steps?

REPLY: We acknowledge that the resolution used for incremental steps between landmarks was indeed large. However, our results show, after 16 cycles of asymmetric rotation, average error was much larger than 45° (additionally, only one of the participants who adapted reported total position error at this minimal threshold; other

errors reported were substantially greater) so the 45° step assures us that adaptation occurred. Below this value, our experience (Pettorossi *et al.* 2013; Panichi *et al.* 2017a, 2017b; Faralli *et al.* 2021) tells us that there was no adaptive effect after several cycles of oscillation. A statement summarising this has been included in the *Experimental Protocol* section of the Methods.

ADDITION (pg 6, line 20):

“Although this resolution was somewhat coarse, previous work (Pettorossi *et al.*, 2013) indicated that 3-4 oscillations usually elicits an error in the range of 30-40°, with additional cycles demonstrating a cumulative error. With participant’s experiencing 16 cycles, it followed that if they didn’t report an error of at least 45°, adaptation had not occurred.”

More importantly, as stated in a response above, the main purpose of collecting and reporting position error was to simply confirm, from a behavioural point of view, that adaptation has occurred. The psychophysics of this adaptive response have already been described in detail in previous work (Pettorossi *et al.* 2013; Panichi *et al.* 2017a, 2017b; Faralli *et al.* 2021); the novelty here lies in the associated EEG markers. More precision may have allowed for us to define a quantitative relationship between EEG activity and the extent of adaptation, but there was concern that additional measuring devices would have introduced electrical interference with the EEG signal, as well as produce motor-related signals that would have been difficult to dissociate from the markers of vestibular adaptation we were interested in. This may be addressed in future studies.

Regarding your comment about when subjects were required to report their perceived orientations, we apologise for the lack of the clarity. Subjects were asked to report their final perceived orientation once per condition at the end of the 16 cycles (and after the two-minute post-adaptation resting state recording). This has been clarified in the Methods section (*Experimental Protocol*).

ADDITION (pg 7, line 1):

“Finally, at the culmination of the recording session, participants were asked to state their final perceived facing.”

2. In the symmetric condition, the errors could be very large for the slow motion condition. For example, as shown in Figure 2A, some subjects show error more than one cycle (>360 deg). In these cases, did the subjects really understand the task, or the errors really reflect their sensory error?

We appreciate your comment regarding subject performance. The larger perceptual error noted in the lower frequency symmetric stimulus was indeed unexpected. However, this may be attributed to that fact that the overall frequency of the stimulus was toward the lower edge of vestibular sensitivity; direction detection thresholds for yaw rotation increase at frequencies of ≤ 0.2 Hz (Grabherr *et al.*, 2008; Carriot *et al.*, 2014; Mallery *et al.*, 2014). As such, it is possible that some participants struggled to discriminate between leftward and rightward motion, and thus perceived themselves to be rotating in one direction. Note that this explanation is distinct from that of the behavioural adaptation that occurs during asymmetric stimulation, which we propose is an effect of attentional reorienting to the more relevant stimulus (i.e., the FHC). This distinction is supported by the differing trends of alpha rhythm resynchronisation during symmetric versus asymmetric stimulation.

ADDITION (pg 15, line 10):

“Firstly, it was noted that there was a considerable degree of error in the low-frequency symmetric condition, although this error was noted in both directions. We attribute this to the stimulus frequency; direction detection thresholds rise below 0.2 Hz rotations (Grabherr *et al.*, 2008; Carriot *et al.*, 2014; Mallery *et al.*, 2014), and as such some participants may have perceived all motion in one direction. This is still behaviourally distinct, however, from the adaptation noted with the asymmetric stimulus, the proposed purpose of which is discussed in greater detail below.”

3. All the figures are not clear enough. They should have higher resolution and quality.

REPLY: Thank you for your comment. We have recreated figures at higher resolution.

Referee #2:

Thank you for submitting your manuscript to The Journal of Physiology.

You must begin the Methods section with the sub-heading "Ethical approval". The relevant information required in this section is already provided by the authors in the opening paragraph.

REPLY: A section titled *Ethical Approval* (pg 4, line 16) has been added to the beginning of the Methods section, in line with journal requirements.

Please refer to "participants" rather than "subjects".

REPLY: All instances of the term "subjects" have been replaced with the term "participants". First instance of this replacement is has been indicated in the revised text (pg 4, line 23); the remaining instances are not highlighted.

It would be helpful to include a statement that there were no adverse outcomes and no lasting disorientation in the participants at the close of the study.

REPLY: A statement to this affect has been added at the end of the *Experimental Protocol* section of methods.

ADDITION (pg 7, line 8): "No adverse outcomes nor persistent sensations of disorientation were reported by participants after study completion."

Referee #3:

In the present manuscript, the authors first replicated previous findings that asymmetric body stimulation (rotation along the z-axis) induces perceptual errors (so-called "vestibular neglect") although it does not impact brainstem reflex mechanisms such as the VOR. Secondly, the authors showed that the alpha frequency band was related to the adaptive process induced by asymmetric rotation. Although the current experiment is of interest and appears to be well conducted, I have some concerns that should be taken into account.

Majors

1) Looking at Figure 1, the traces of the samples reflect the fact that the chair is manually operated, as the profiles of these traces do not correspond to pure sinusoids, which is particularly visible with regard to the velocity. The fact that the chair is manually operated is understandable if one considers that the authors wish to limit the electrical noise in a Faraday cage. However, if one considers that the vestibular system (here the semi-circular canals) is sensitive to accelerations, one can assume that the acceleration profiles are even further away from pure sinusoids. Considering that the traces reported in the figure are probably among the most presentable traces, can the authors reassure the reader about the quality of the stimulus considering that the experimental aim is to evaluate a perceptual dimension (vestibular neglect) that arises from an acceleration

sensitive system.

REPLY: Thank you for your comment. Stimuli were hand delivered always by the same experimenter (the first author) who is an ex-professional dancer with ample experience in rhythmic, repetitive movements. Further, recall the manual oscillation was guided by a metronome and an amplitude calibrated visual display. We instrumented a training period until we were satisfied that the rotations were as intended before starting the experiments. Once the experiments commenced, traces were always visually inspected at the end of each session to ensure consistency and regularity of the stimulus profile. In addition, at the end of the project we randomly selected 10 participants and overlaid their stimuli to visualise discrepancies between traces. We also performed a Fast Fourier Transform (FFT) of the stimulus traces, overlaying the FFT results as well. These data are shown in **Supplementary Figures 1 and 2**, respectively, showing minimal variability in the shape and frequency content of the stimuli. A statement to the above effect was added to the *Experimental Design* section of the Methods, as stated below.

ADDITION (pg 5, line 28): “The experimenter practised rotating the chair at these frequencies at length prior to beginning the experiment; traces were examined at the end of each session to ensure regularity and consistency of the stimulus. In addition, at the conclusion of the study, the recorded stimulus traces of 10 randomly selected participants were overlaid to check for discrepancies (**Supplementary Figure 1**). A Fast Fourier Transform was also performed on the traces, and the results overlaid (**Supplementary Figure 2**). Both analyses showed minimal variability in the shape and frequency content of the stimuli, thus demonstrating the stimulation was consistent across participants.”

2) The authors report a dissociation between the VOR, which is unaffected by stimulus asymmetry, which is also a direct measure of vestibular function, and perception, which is particularly sensitive to this asymmetry and is thought to reflect a high level of vestibular interpretation. It turns out that the rotation of the participants to the right and to the left stimulates of course the vestibular system, but does also stimulate the proprioceptive system (through pressures on the body surface that differ according to the speed and acceleration of the body in space). In this respect, one may wonder whether the dissociation between VOR and perception (consider as vestibular neglect) in response to stimulus asymmetry does not only vestibular signals involvement in reflex activities (VOR) but vestibular as well as proprioceptive signals for perception. This is should definitely be discussed.

REPLY: We thank you for highlighting a potential confound in isolating vestibular input. A small proprioceptive component during asymmetric rotation signal may be elicited. However, given the transfer characteristics of the vestibular and proprioceptive systems it would be unlikely that any putative proprioceptive input would have a significant influence on our asymmetric adaptation paradigm. In a study by Mergner et al (2001) and in my own (Nakamura and Bronstein, 1995; e.g., see Figure 6 there) the gain of the vestibulo-perceptual response increases as a function of stimulus frequency/velocity, in contrast to the gain of the proprioceptive perceptual response which remains essentially unchanged. This means that any increased proprioceptive input brought about by the higher acceleration present during the fast hemicycle would be relatively less efficiently sensed by the proprioceptive than by the vestibular system. Additionally, asymmetrical body rotations with passive head rotation were previously examined to look for the role of neck proprioception on movement perception and found to not impact the perceptual adaptation (Panichi *et al.*, 2011). In addition, recent published articles examined this paradigm in vestibular neuritis (Panichi *et al.*, 2017a) and Meniere's disease (Faralli *et al.*, 2021) patient populations, showing that vestibular impairment significantly alters this adaptive perceptual response. Thus, the vestibular system is the system most involved in the adaptive phenomenon. This has been added to the *Potential Role of Subcortical Circuits* section of the Discussion.

ADDITION (pg 19, line 27):

“A separate consideration is the possible influence of low-level proprioceptive signals on the adaptive response. Although every measure was taken to restrict proprioceptive input during the experimental set-up, it is possible there was a proprioceptive contribution to the perceptual response. However, it is unlikely that this had a significant effect on the adaptation noted. Increased stimulus frequency has been shown to increase the gain of the vestibulo-perceptual response, but not the proprioceptive response (Nakamura & Bronstein, 1995; Mergner *et al.*, 2001), meaning the increased acceleration of the FHC would be sensed less efficiently by the proprioceptive system relative to the vestibular system. Further evidence of this is provided by previous work which found that passive head deviations during asymmetric rotations had no impact on movement perception (Panichi *et al.*, 2011). Additionally, studies of this paradigm in vestibular neuritis (Panichi *et al.*, 2017a) and Meniere's disease (Faralli *et al.*, 2021) populations have shown that vestibular impairment alters the adaptive response noted in healthy individuals. Taken together, these findings, along with those in the present study, support that the adaptation is due to altered vestibular perception.”

3) The definition of alpha rhythm 8-14 Hz is atypical and does not fit to the recommended standard (please see table 1 of Babiloni et al. 2020 for a recent review),

which is important for comparing studies. The upper bound is in most cases equal to 12 Hz and should never be higher than 13 Hz. The authors should define the alpha bandwidth in respect of these international guidelines, or at least discuss the reasons of this discrepancy.

REPLY: We acknowledge that the upper range of alpha frequency is often defined at 12 or 13 Hz, particularly when averaging across an entire band. Alpha power used for our statistical analyses, however, was taken from the individual alpha frequency (IAF) for each participant. This measure has historically been determined as the peak between 6 and 14 Hz (Klimesch, 1999) and studies that utilise IAF-defined bands have extended the upper limit to 14 Hz (Babiloni *et al.*, 2020b). This has been clarified in the *EEG Analysis* section of the Methods.

ADDITION (pg 8, line 6):

“For the alpha frequency band, power was measured at individual alpha frequency (IAF). The upper bound of alpha frequencies is usually defined at 12 or 13 Hz; however, IAF has been found to go up to 14Hz (Klimesch, 1999; Babiloni *et al.*, 2020b, 2020a), hence the larger band.”

That said, the average IAF across all participants (averaged across all electrodes) was 10.47 Hz ($SD = 0.85$ Hz), with the minimum IAF measured at 8.43 Hz and the maximum at 12.68 Hz. These descriptive statistics have been added to the *Time-Based Spectral Analysis* section of the results.

ADDITION (pg 12, line 31):

“Individual alpha frequency (IAF) – the point at which power was measured – averaged (SD) at 10.47 Hz (0.85 Hz) across all regions, with values ranging from 8.43 to 12.68 Hz.”

4) The Time-Based Spectral Analysis section begins with several visual observations and (what seems to be) arbitrary choices. First, the authors stated that "Average normalised power of each cycle was also examined and confirmed the previously noted observations, namely that there was a greater suppression of alpha power in the symmetric condition relative to the asymmetric condition and that the parietal, occipital, prefrontal, and frontal regions showed this response more clearly (Figure 3B). There was no observable lateralisation of this response." The authors should run statistical analyses in order to support these observations (i.e., sign. difference from baseline, lateralization), prior to the ROI creation ("As a result, power was averaged across the electrodes in these four regions, henceforth referred to as regions of interest (ROIs), for further analysis »). Second, the authors should justify the choice for plotting O2 for Fig 3A. Also, why not plot the last cycle in Fig 3A?

REPLY: Thank you for asking for additional clarification. We viewed the visual observations as an integral part of the data analysis process. We trust that the Referee agrees that visual examination of raw physiological data has always been utilised to get a sense of possible trends in the data and inform the direction of statistical analysis.

ADDITION (pg 12, line 6):

“Close visual examination of spectrograms at various electrodes was utilised to roughly identify trends in the data. This preliminary analysis yielded several important observations.”

Part of the reason for choosing to use the spectrograms of the O2 electrode in the manuscript is that it demonstrates one of our initial visual observations quite clearly, namely that alpha power (which is very visible as the bright yellow band in the spectrograms, particularly in occipital electrodes) was greater at baseline than during the adaptation period, for both the symmetric and asymmetric conditions.

ADDITION (pg 12, line 11):

“This can be clearly observed in the occipital region, which is known to receive substantial propagation of alpha rhythms from subcortical generators (Halgren *et al.*, 2019); the O2 electrode, depicted in **Figure 3A**, demonstrates this well.”

Regarding the comparison between baseline measures and those during the *Adaptation* period, the analysis of the time-based spectral data uses normalised power values, i.e., raw power (averaged across an epoch; the epoch corresponding to one full oscillation of the chair) during the *Adaptation* period divided by raw power (averaged across a corresponding epoch) during the *Baseline* period. This normalisation served two purposes; first it reduces the high degree of variance in raw power values both within and between subjects. Second, it directly quantifies the relationship between power at baseline versus power during adaptation, i.e., a value less than 1 would indicate power during the *Adaptation* period is less than power at baseline (also termed here as desynchronisation), while a value greater than 1 indicates power is greater than at baseline (increased synchronisation). The relationship between power and synchronicity is described in greater detail by Pfurtscheller and colleagues (Pfurtscheller & Lopes Da Silva, 1999; Pfurtscheller, 2001). We used a simplified version of the calculations specified there. As such, the signal difference from baseline is already included in the statistical model for all spectral analysis (time-based and steady state).

ADDITION (pg 8, line 10):

“To reduce inter-individual variance of EEG, the *Adaptation* and *Post-Adaptation* recordings were normalised (divided) by the baseline recording. This normalisation

process additionally provided an indication of the relationship between raw power during the *Adaptation* period versus the *Baseline* period. Specifically, it indicates if power is decreased during the *Adaptation* period relative to baseline, (note: as this reduction results from decreased synchrony of the underlying neural population, it is termed desynchronisation) or increased (increased synchrony; termed synchronisation) (Pfurtscheller & Lopes Da Silva, 1999; Pfurtscheller, 2001). Thus, normalised power values greater than 1 indicated synchronisation, values less than 1 indicated desynchronisation.”

Selection of the ROIs was largely determined *a priori* from initial literature review, which, as summarised in the introduction, highlighted the role of the parieto-occipital cortices in vestibular processing (Lopez & Blanke, 2011; Gale *et al.*, 2016) as well as the role of frontoparietal attentional networks in error processing and sensory adaptation (Corbetta & Shulman, 2002; Petersen & Posner, 2012; Scolari *et al.*, 2015). Selection of ROIs was additionally determined from our initial visual observation of the averaged spectrogram data (i.e., the headplots shown in **Figure 3B**). Rationale for ROI selection and the list of electrodes included in each ROI have been specified in the *Time-Based Spectral Analysis* section of the Results.

ADDITION (pg 12, line 19):

“Regions of interest were partially selected *a priori*, based on previous work postulating the role of the parieto-occipital cortices in vestibular processing (Lopez & Blanke, 2011; Gale *et al.*, 2016) as well as the role of frontoparietal attentional networks in error processing and sensory adaptation (Corbetta & Shulman, 2002; Petersen & Posner, 2012; Scolari *et al.*, 2015). The decision to focus on these regions was further confirmed through visual examination of the average normalised power of each spectrogram (**Figure 3B**), which aligned the previously noted observations; there was a greater suppression of alpha power in the symmetric condition relative to the asymmetric condition and this was largely reflected in the parietal (P7, P3, Pz, P4, and P8 electrodes), occipital (O1, Oz, and O2 electrodes), prefrontal (Fp1, Fpz, and Fp2 electrodes), and frontal (F7, F3, Fz, F4, and F8 electrodes) regions (**Figure 3B**).”

The last cycle has been plotted in Figure 3A.

5) The same goes for the ROI definition of Steady state spectral power analysis: "Head plots of steady state power showed no lateralisation either the Adaptation or Post-Adaptation in any frequency band, but several regions appeared to differ between conditions, including frontal, prefrontal, parietal, centroparietal, and occipital electrodes (Figure 5A). These electrodes were grouped by region ». Also, are these ROI different from the previous ones? Please clarify for the reader.

REPLY: We apologise for the lack of clarity around the definitions of ROIs. The ROIs used during the steady state analysis are identical to those described earlier and were chosen in an identical manner to that described in the previous comment. The only distinction is that centroparietal electrodes, which were observed to differ between conditions, were added as a ROI. This has been clarified in the *Steady State Spectral Power* section of the Results.

ADDITION (pg 13, line 26):

“As in the previous section, visual examination was conducted on normalised steady state power head plots to determine ROIs to analyse. No lateralisation was observed in either the *Adaptation* or *Post-Adaptation* in any frequency band, but several regions (most of which were predicted to be relevant *a priori*) appeared to differ between conditions, including frontal, prefrontal, parietal, centroparietal, and occipital electrodes (**Figure 5A**). These electrodes were grouped by region (same ROIs as before, but with the centroparietal electrodes – CP5, CP1, CP2, AND CP6 – being included)”

6) Even though they are of great interest, I find the results and interpretations of the functional connectivity analysis rather ambitious in the space domain, given the fact that it is based on average magnitude-squared coherence. In scalp (referenced) EEG, such measurement is very prone to artefacts from volume conduction. I would recommend reprocessing data using the imaginary part of coherence for scalp level analysis (see Nolte et al., 2004), or applying spatial filtering (such as Laplacian) prior to coherence computation or going into source space analysis (even in absence of individual MRI, i.e., on a brain template). Also, the authors should show the connectivity pattern at rest, i.e., during baseline (it is stated in the methods, but no results are discussed afterwards - or is it included somehow in the statistical model? Please clarify).

REPLY: We acknowledge that volume conduction is typically an issue when conducting coherence analysis. This is more problematic, however, when conducting experiments with a between-subjects design (i.e., different subject groups). As we were comparing the influence of different stimuli on the same group of participants (i.e., a within-subjects design), factors that usually influence volume conduction, such as tissue thickness, are inherently controlled by the study design.

Another factor usually related to volume conduction is proximity of electrodes; however, the edges (connections) identified in our network analysis are mostly comprised of distant nodes (electrodes), namely (pre)frontal and parieto-occipital electrodes. Furthermore, the electrodes between these regions do not show significant connectivity (except for CP1, CPz, and Pz in the beta frequency band; this will be discussed momentarily) and connections are fairly lateralised; if volume conduction was driving the

results, we would expect to see multiple connections between neighbouring electrodes on both sides of the scalp. Given this, we argue that our results are valid and are largely indicative of stimulus-related changes in network connectivity.

That said, we did reprocess data using imaginary coherence (IC) to examine the cluster of neighbouring electrodes mentioned above, which may have been more influenced by volume conduction. The reanalysis found a significant difference in network strength between conditions during the adaptation period for beta frequency ($p = 0.039$, see Figure 6B). Although the connections identified differed from those identified in the previous analysis of magnitude-squared coherence (MSC), more than 50% of the electrodes in the MSC-derived network were also involved in the IC-derived network. Most interestingly, inter-electrode connectivity findings were reproduced for the same cluster of midline “z” electrodes. While unexpected to find this pattern maintained, there is strong evidence that midline electrodes pick up late vestibular activity (Todd *et al.*, 2014; Nakul *et al.*, 2021). The above has been added to the results and discussion where applicable.

ADDITION (pg 14, line 22):

“Concern around the results being attributable to volume conduction (VC) was minimal, given that we utilised a within-subjects design (i.e., many factors influencing VC were inherently controlled for), and given the distance between nodes (electrodes) comprising edges (connections). However, there was a cluster of neighbouring electrodes identified in the beta network (CP1, CPz, and Pz) that were identified as potentially susceptible to VC. To further investigate, data were reprocessed using imaginary coherence (Nolte *et al.*, 2004). This analysis identified a network of 12 connections in the beta band only ($FWER-p = 0.039$), primarily between bilateral frontocentral and centroparietal electrodes (**Figure 6B**).”

ADDITION (pg 17, line 30):

“The results of the imaginary coherence analysis were unexpected, as the network identified was comprised almost exclusively of fronto-central and centroparietal connections. However, it has been theorised that midline regions can reflect late latency vestibular signals (Todd *et al.*, 2014; Nakul *et al.*, 2021), which may reflect some of the bottom-up feed-forward mechanisms discussed previously.”

Finally, regarding networks at baseline, thank you for identifying the lack of clarity in the methods. The comparison made in these analyses is between the asymmetric and symmetric condition (which serves as a control); this comparison was run *separately* for the baseline, adaptation, and post-adaptation periods, as these were all distinct states

(i.e., we would not expect to find the same network at baseline as we would at during specific task). This has been clarified in the *statistical Analysis* section of the Methods.

ADDITION (pg 10, line 21):

“Average magnitude-squared coherence was used as the measure of connectivity for our networks, which directly compared the asymmetric and symmetric conditions, separately for each period (*Baseline, Adaptation, and Post-Adaptation*).

7) The authors consider that the temporary vestibulo-perceptual "neglect" induced by asymmetric vestibular stimulation could be mediated by alpha rhythms and frontoparietal attentional networks. This is consistent with previous data showing that vestibular processing as well as self-motion perception are related to modulation of alpha power over the left and right temporoparietal regions (Gale et al., 2016 [cited] but see and cite also Gutteling & Medendorp, 2016). This should also be discussed in relation to recent research showing an increase in alpha activity when self-motion perception is visually induced -vection- (Harquel et al., 2020; McAssey et al., 2020). This is of particular interest as the level of alpha activity in the vestibular cortical area varies with visually induced self-motion. These later results deserve to be put into perspective with the vestibule-perceptual "neglect" vs EEG modulation reported here. REPLY: Thank you for suggesting these papers. They indeed align with our results nicely. They have been incorporated in the Discussion as follows

ADDITION (pg 15, line 31):

“...specifically that desynchronisation and subsequent rebound of alpha power differs in response to asymmetric versus symmetric stimulation, particularly in the prefrontal and occipital regions, where the resynchronisation is accelerated and diminished, respectively. Body motion has been shown to correspond with a reduction of alpha power in central parietal, and lateral parieto-temporal areas (Gutteling & Medendorp, 2016). Gale *et al.* (2016) additionally showed that a continuously perceived vestibular stimulation is associated with suppression of alpha over the parietal cortex. Our findings show the same initial reduction, which then decreases over the course of continual stimulation. This trend has been noted in the literature previously; a relative increase after initial reduction was noted by Gale et al. (2016), as well as in separate studies usingvection to induce motion perception (Harquel *et al.*, 2020; McAssey *et al.*, 2020).

Our findings, along with previous work, suggest that the initial alpha reduction indicates neural pathways are primed to detect and process incoming information, but then are fine-tuned to process salient information. This supposition, which is also discussed in similar work (Pettorossi *et al.*, 2013; Harquel *et al.*, 2020) is supported by work on the role of enhanced alpha as an inhibitory gating mechanism...”

Minors

8) Please specify what was the a priori rule for defining nystagmus in EOG data.

REPLY: Vestibular nystagmus was identified when a consistent slow phase eye movement of at least 4 deg/s was followed by an oppositely directed fast phase. A sentence clarifying this criterion has been added to the *Statistical Analysis* section of the Methods.

ADDITION (pg 9, line 17):

“Nystagmus was identified by trained experimenters (first, third, and sixth authors) as characteristic jerk nystagmus (Serra & Leigh, 2002), i.e., slow drift of at least 4°/s from a fixation point immediately followed by rapid corrective motion in the opposite direction. Traces that did not demonstrate clear nystagmus were rejected.”

9) The description of the EEG preprocessing and analysis is lacking some details. Please specify the average +/- std length of the resulting epochs (is there a difference between cycles? This is important in respect of spectral analysis like FFT). Please specify ROI by depicting electrodes defining each ROI (on a blank topoplot figure for example).

REPLY: We apologise for the lack of clarification regarding epoch length. Epochs were all identical in length, measured as 5 seconds (asymmetric) or 8 seconds (symmetric) from peak displacement of the chair. A sentence clarifying this has been added to the Methods.

ADDITION (pg 8, line 21):

“Epochs were 5s (asymmetric) or 8s (symmetric) in length, measured from peak displacement of each cycle of the chair position traces.”

A blank topoplot with regions identified was added to Figure 4.

10) The Figure 1Cii referenced in page 7 is missing.

REPLY: Thank you for identifying this error. The erroneous reference to a non-existent figure has been removed.

11) For clarity, the authors should not overlay electrodes positions using black dots in topographic views, since it is hiding the electrical field (and given the fact that the EEG cap is in the 10-20 standard system).

REPLY: Thank you for this suggestion. Headplots have been reformatted without the electrode positions and replaced in the applicable figures.

Dear Dr Bronstein,

Re: JP-RP-2022-282470R2 "Electroencephalographic Response to Transient Adaptation of Vestibular Perception" by Josephine I Cooke, Onur Guven, Patricia Castro Abarca, Richard T Ibitoye, Vito E Pettorossi, and Adolfo M. Bronstein

I am pleased to tell you that your paper has been accepted for publication in The Journal of Physiology.

NEW POLICY: In order to improve the transparency of its peer review process The Journal of Physiology publishes online as supporting information the peer review history of all articles accepted for publication. Readers will have access to decision letters, including all Editors' comments and referee reports, for each version of the manuscript and any author responses to peer review comments. Referees can decide whether or not they wish to be named on the peer review history document.

The last Word version of the paper submitted will be used by the Production Editors to prepare your proof. When this is ready you will receive an email containing a link to Wiley's Online Proofing System. The proof should be checked and corrected as quickly as possible.

Authors should note that it is too late at this point to offer corrections prior to proofing. The accepted version will be published online, ahead of the copy edited and typeset version being made available. Major corrections at proof stage, such as changes to figures, will be referred to the Reviewing Editor for approval before they can be incorporated. Only minor changes, such as to style and consistency, should be made a proof stage. Changes that need to be made after proof stage will usually require a formal correction notice.

All queries at proof stage should be sent to TJP@wiley.com

Are you on Twitter? Once your paper is online, why not share your achievement with your followers. Please tag The Journal (@jphysiol) in any tweets and we will share your accepted paper with our 23,000+ followers!

Yours sincerely,

Richard Carson
Senior Editor
The Journal of Physiology

P.S. - You can help your research get the attention it deserves! Check out Wiley's free Promotion Guide for best-practice recommendations for promoting your work at www.wileyauthors.com/eeo/guide. And learn more about Wiley Editing Services which offers professional video, design, and writing services to create shareable video abstracts, infographics, conference posters, lay summaries, and research news stories for your research at www.wileyauthors.com/eeo/promotion.

*** IMPORTANT NOTICE ABOUT OPEN ACCESS ***

To assist authors whose funding agencies mandate public access to published research findings sooner than 12 months after publication The Journal of Physiology allows authors to pay an open access (OA) fee to have their papers made freely available immediately on publication.

You will receive an email from Wiley with details on how to register or log-in to Wiley Authors Services where you will be able to place an OnlineOpen order.

You can check if your funder or institution has a Wiley Open Access Account here <https://authorservices.wiley.com/author-resources/Journal-Authors/licensing-and-open-access/open-access/author-compliance-tool.html>

Your article will be made Open Access upon publication, or as soon as payment is received.

If you wish to put your paper on an OA website such as PMC or UKPMC or your institutional repository within 12 months of publication you must pay the open access fee, which covers the cost of publication.

OnlineOpen articles are deposited in PubMed Central (PMC) and PMC mirror sites. Authors of OnlineOpen articles are permitted to post the final, published PDF of their article on a website, institutional repository, or other free public server, immediately on publication.

Note to NIH-funded authors: The Journal of Physiology is published on PMC 12 months after publication, NIH-funded authors DO NOT NEED to pay to publish and DO NOT NEED to post their accepted papers on PMC.

EDITOR COMMENTS

Reviewing Editor:

I have no further comments.